# A new model of decision processing in instrumental learning tasks

**Steven Miletić[1]\*, Russell J Boag[1], Anne C Trutti[1,2], Niek Stevenson[1], Birte U Forstmann[1], Andrew Heathcote[1,3]**

[1]University of Amsterdam, Department of Psychology, Amsterdam, Netherlands; [2]Leiden University, Department of Psychology, Leiden, Netherlands; [3]University of Newcastle, School of Psychology, Newcastle, Australia

**Abstract** Learning and decision-making are interactive processes, yet cognitive modeling of error-driven learning and decision-making have largely evolved separately. Recently, evidence accumulation models (EAMs) of decision-making and reinforcement learning (RL) models of error-driven learning have been combined into joint RL-EAMs that can in principle address these interactions. However, we show that the most commonly used combination, based on the diffusion decision model (DDM) for binary choice, consistently fails to capture crucial aspects of response times observed during reinforcement learning. We propose a new RL-EAM based on an advantage racing diffusion (ARD) framework for choices among two or more options that not only addresses this problem but captures stimulus difficulty, speed-accuracy trade-off, and stimulus-response-mapping reversal effects. The RL-ARD avoids fundamental limitations imposed by the DDM on addressing effects of absolute values of choices, as well as extensions beyond binary choice, and provides a computationally tractable basis for wider applications.

## Introduction

Learning and decision-making are mutually influential cognitive processes. Learning processes refine the internal preferences and representations that inform decisions, and the outcomes of decisions underpin feedback-driven learning (*Bogacz and Larsen, 2011*). Although this relation between learning and decision-making has been acknowledged (*Bogacz and Larsen, 2011*; *Dayan and Daw, 2008*), the study of cognitive processes underlying feedback-driven learning on the one hand, and of perceptual and value-based decision-making on the other, have progressed as largely separate scientific fields. In the study of error-driven learning (*O'Doherty et al., 2017*; *Sutton and Barto, 2018*), the decision process is typically simplified to soft-max, a descriptive model that offers no process-level understanding of how decisions arise from representations, and ignores choice response times (RTs). In contrast, evidence-accumulation models (EAMs; *Donkin and Brown, 2018*; *Forstmann et al., 2016*; *Ratcliff et al., 2016*) provide a detailed process account of decision-making but are typically applied to tasks that minimize the influence of learning, and residual variability caused by learning is treated as noise.

Recent advances have emphasized how both modeling traditions can be combined in joint models of reinforcement learning (RL) and evidence-accumulation decision-making processes, providing mutual benefits for both fields (*Fontanesi et al., 2019a*; *Fontanesi et al., 2019b*; *Luzardo et al., 2017*; *McDougle and Collins, 2020*; *Miletić et al., 2020*; *Millner et al., 2018*; *Pedersen et al., 2017*; *Pedersen and Frank, 2020*; *Sewell et al., 2019*; *Sewell and Stallman, 2020*; *Shahar et al., 2019*; *Turner, 2019*). Combined models generally propose that value-based decision-making and learning interact as follows: For each decision, a subject gradually accumulates evidence for each choice option by sampling from a running average of the subjective value (or *expected reward*) associated with each choice option (known as *Q-values*). Once a threshold level of evidence is reached,

**\*For correspondence:**
s.miletic@uva.nl

they commit to the decision and initiate a corresponding motor process. The response triggers feedback, which is used to update the internal representation of subjective values. The next time the subject encounters the same choice options this updated internal representation changes evidence accumulation.

The RL-EAM framework has many benefits (*Miletić et al., 2020*). It allows for studying a rich set of behavioral data simultaneously, including entire RT distributions and trial-by-trial dependencies in choices and RTs. It posits a theory of evidence accumulation that assumes a memory representation of rewards is the source of evidence, and it formalizes how these memory representations change due to learning. It complements earlier work connecting theories of reinforcement learning and decision-making (*Bogacz and Larsen, 2011*; *Dayan and Daw, 2008*) and their potential neural implementation in basal ganglia circuits (*Bogacz and Larsen, 2011*), by presenting a measurement model that can be fit to, and makes predictions about, behavioral data. Adding to benefits in terms of theory building, the RL-EAM framework also has potential to improve parameter recovery properties compared to standard RL models (*Shahar et al., 2019*), and allows for the estimation of single-trial parameters of the decision model, which can be crucial in the analysis of neuroimaging data.

An important challenge of this framework is the number of modeling options in both the fields of reinforcement learning and decision-making. Even considering only model-free (as opposed to model-based, see *Daw and Dayan, 2014*) reinforcement learning, there exists a variety of learning rules (e.g. *Palminteri et al., 2015*; *Rescorla and Wagner, 1972*; *Rummery and Niranjan, 1994*; *Sutton, 1988*), as well as the possibility of multiple learning rates for positive and negative prediction errors (*Christakou et al., 2013*; *Daw et al., 2002*; *Frank et al., 2009*; *Gershman, 2015*; *Frank et al., 2007*; *Niv et al., 2012*), and many additional concepts, such as eligibility traces to allow for updating of previously visited states (*Barto et al., 1981*; *Bogacz et al., 2007*). Similarly, in the decision-making literature, there exists a wide range of evidence-accumulation models, including most prominently the diffusion decision model (DDM; *Ratcliff, 1978*; *Ratcliff et al., 2016*) and race models such as the linear ballistic accumulator model (LBA; *Brown and Heathcote, 2008*) and racing diffusion (RD) models (*Boucher et al., 2007*; *Hawkins and Heathcote, 2020*; *Leite and Ratcliff, 2010*; *Logan et al., 2014*; *Purcell et al., 2010*; *Ratcliff et al., 2011*; *Tillman et al., 2020*).

The existence of this wide variety of modeling options is a double-edged sword. On the one hand, it highlights the success of the general principles underlying both modeling traditions (i.e. learning from prediction errors and accumulate-to-threshold decisions) in explaining behavior, and it allows for studying specific learning/decision-making phenomena. On the other hand, it constitutes a bewildering combinatorial explosion of potential RL-EAMs; here, we provide empirical grounds to navigate this problem with respect to EAMs.

The DDM is the dominant EAM as currently used in reinforcement learning (*Fontanesi et al., 2019a*; *Fontanesi et al., 2019b*; *Millner et al., 2018*; *Pedersen et al., 2017*; *Pedersen and Frank, 2020*; *Sewell et al., 2019*; *Sewell and Stallman, 2020*; *Shahar et al., 2019*), but this choice is without experimental justification. Furthermore, the DDM has several theoretical drawbacks, such as its inability to explain multi-alternative decision-making and its strong commitment to the accumulation of the evidence *difference*, which leads to difficulties in explaining behavioral effects of absolute stimulus and reward magnitudes without additional mechanisms (*Fontanesi et al., 2019a*; *Ratcliff et al., 2018*; *Teodorescu et al., 2016*). Here, we compare the performance of different decision-making models in explaining choice behavior in a variety of instrumental learning tasks. Models that fail to capture crucial aspects of performance run the risk of producing misleading psychological inferences. For EAMs, the full RT distribution (i.e. its level of variability and skew) have proven to be crucial. Hence, it is important to assess which RL-EAMs are able to capture not only learning-related changes in choice probabilities and mean RT, but also the general shape of the entire RT distribution and how it changes with learning. Further, in order to be held forth as a general modeling framework, it is important to capture how all these measures interact with key phenomena in the decision-making and learning literature.

We compare the RL-DDM with two RL-EAMs based on a racing accumulator architecture (*Figure 1*). All the RL-EAMs assume evidence accumulation is driven by Q-values, which change based on error-driven learning as governed by the classical delta update rule. Rather than a two-sided DDM process (*Figure 1A*), the alternative models adopt a neurally plausible RD architecture (*Ratcliff et al., 2007*), which conceptualize decision-making as a statistically independent race between single-sided diffusive accumulators, each collecting evidence for a different choice option.

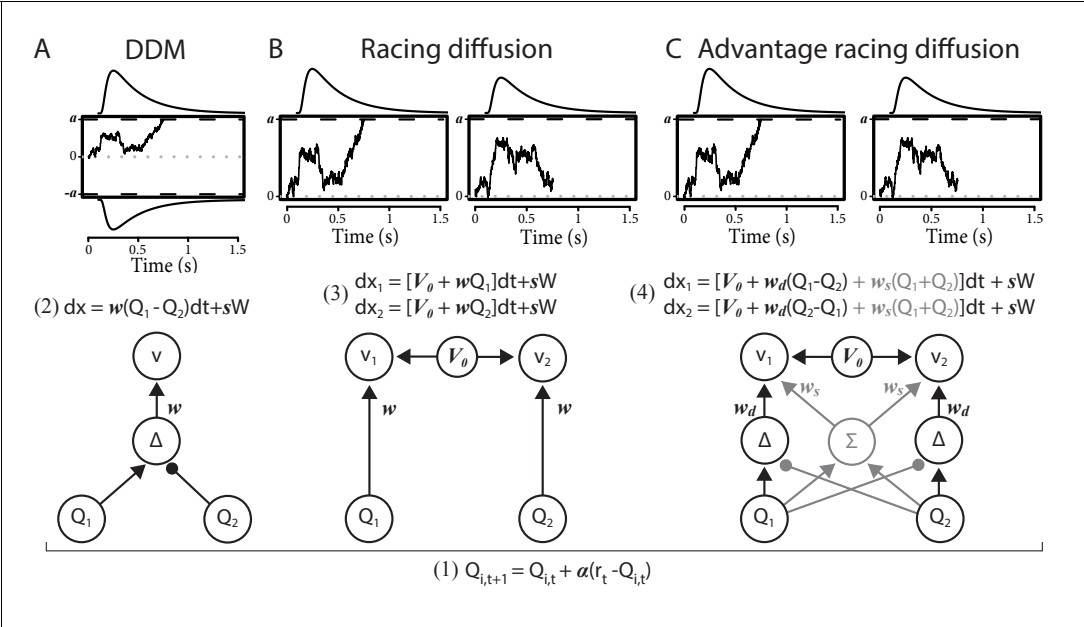

**Figure 1.** Comparison of the decision-making models. Bottom graphs visualize how Q-values are linked to accumulation rates. Top panel illustrates the evidence-accumulation process of the DDM (panel **A**) and racing diffusion (RD) models (panels **B** and **C**). Note that in the race models there is no lower bound. *Equations 2–4* formally link Q-values to evidence-accumulation rates. In the RL-DDM, the difference ($\Delta$) in Q-values is accumulated, weighted by free parameter $w$, plus additive within-trial white noise W with standard deviation $s$. In the RL-RD, the (weighted) Q-values for both choice options are independently accumulated. An evidence-independent baseline urgency term, $V_0$ (equal for all accumulators), further drives evidence accumulation. In the RL-ARD models, the advantages ($\Delta$) in Q-values are accumulated as well, plus the evidence-independent baseline term $V_0$. The gray icons indicate the influence of the Q-value *sum* ($\Sigma$) on evidence accumulation, which is not included in the limited variant of the RL-ARD. In all panels, bold-italic faced characters indicate parameters. $Q_1$ and $Q_2$ are Q-values for both choice options, which are updated according to a delta learning rule (*Equation 1* at the bottom of the graph), with learning rate $\alpha$.

The first accumulator to reach its threshold triggers motor processes that execute the corresponding decision. The alternative models differ in how the mean values of evidence are constituted. The first model, the RL-RD (*Figure 1B*), postulates accumulators are driven by the expected reward for their choice, plus a stimulus-independent baseline (c.f. an *urgency* signal; *Miletić and van Maanen, 2019*). The second model, the RL-ARD (advantage racing diffusion), uses the recently proposed *advantage* framework (*van Ravenzwaaij et al., 2020*), assuming that each accumulator is driven by weighted combination of three terms: the *difference* ('advantage') in mean reward expectancy of one choice option over the other, the *sum* of the mean reward expectancies, and the urgency signal. In perceptual choice, the advantage term consistently dominates the sum term by an order of magnitude (*van Ravenzwaaij et al., 2020*), but the sum term is necessary to explain the effects of absolute stimulus magnitude. We also fit a limited version of this model, RL-lARD, with the weight of the sum term set to zero to test whether accounting for the influence of the sum is necessary even when reward magnitude is not manipulated, as was the case in our first experiments. The importance of sum and advantage terms is also quantified by their weights as estimated in full RL-ARD model fits.

For all models, we first test how well they account for RT distributions (central tendency, variability, and skewness of RTs), accuracies, and learning-related changes in RT distributions and accuracies in a typical instrumental learning task (*Frank et al., 2004*). In this experiment, we also manipulated difficulty, that is, the magnitude of the difference in average reward between pairs of options. In two further experiments, we test the ability of the RL-EAMs to capture key behavioral phenomena in the decision-making and reinforcement-learning literatures, respectively, speed-accuracy trade-off (SAT), and reversals in reward contingencies, again in binary choice. In a final experiment, we show that the RL-ARD extends beyond binary choice, successfully explaining accuracy and full RT distributions from a three-alternative instrumental learning task that manipulates reward magnitude.

# Results

In the first experiment, participants made decisions between four sets of two abstract choice stimuli, each associated with a fixed reward probability (*Figure 2A*). On each trial, one choice option always had a higher expected reward than the other; we refer to this choice as the 'correct' choice. After each choice, participants received feedback in the form of points. Reward probabilities, and therefore choice difficulty, differed between the four sets (*Figure 2B*). In total, data from 55 subjects were included in the analysis, each performing 208 trials (see Materials and methods).

Throughout, we summarize RT distributions by calculating the 10th, 50th (median), and 90th percentiles separately for correct and error responses. The median summarizes central tendency, the difference between 10th and 90th percentiles summarizes variability and the larger difference between the 90th and 50th percentiles than between the 50th and 10th percentiles summarizes the positive skew that is always observed in RT distributions. To visualize the effect of learning, we divided all trials in 10 bins (approximately 20 trials each), and calculated accuracy and the RT percentiles per bin. Note that model fitting was not based on these data summaries. Instead, we used hierarchical Bayesian methods to fit models to the data from every trial and participant

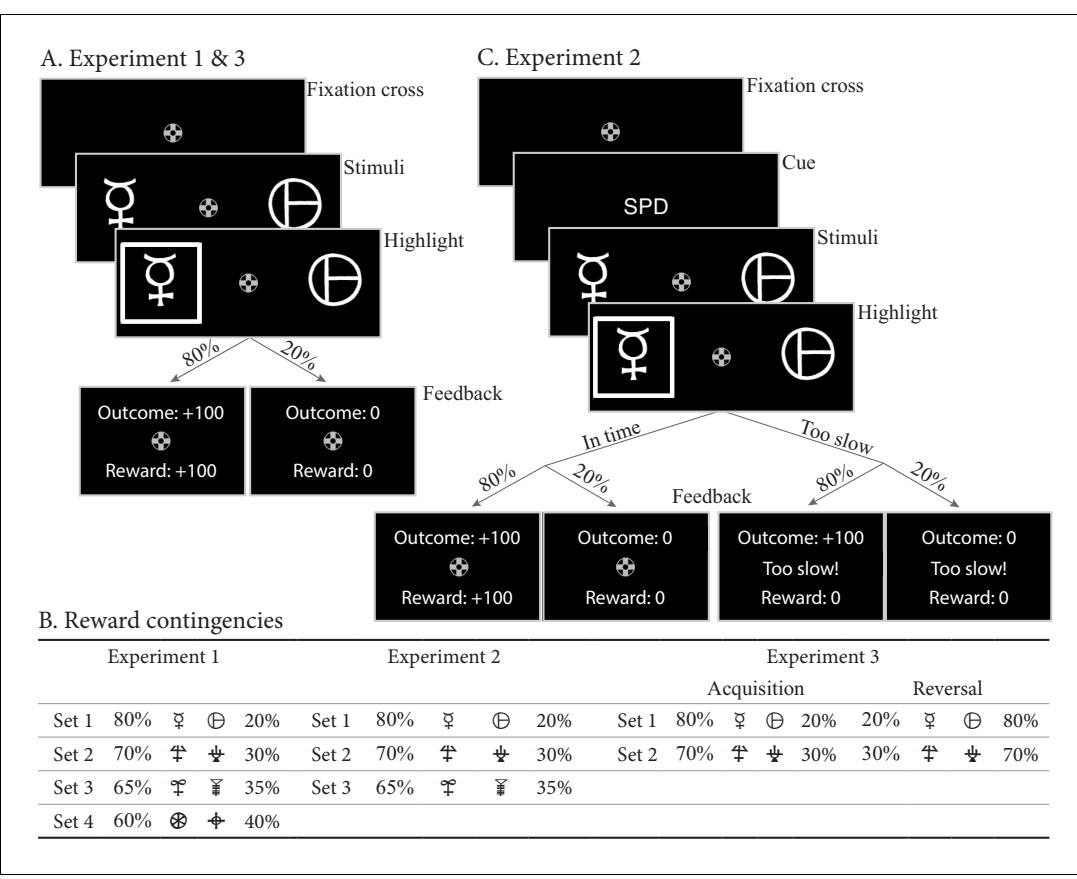

**Figure 2.** Paradigms for experiments 1–3. (**A**) Example trial for experiments 1 and 3. Each trial starts with a fixation cross, followed by the presentation of the stimulus (until choice is made or 2.5 s elapses), a brief highlight of the chosen option, and probabilistic feedback. Reward probabilities are summarized in (**B**). Percentages indicate the probabilities of receiving +100 points for a choice (with 0 otherwise). The actual symbols used differed between experiments and participants. In experiment 3, the acquisition phase lasted 61–68 trials (uniformly sampled each block), after which the reward contingencies for each stimulus set reversed. (**C**) Example trial for experiment 2, which added a cue prior to each trial ('SPD' or 'ACC'), and had feedback contingent on both the choice and choice timing. In the SPD condition, RTs under 600 ms were considered in time, and too slow otherwise. In the ACC condition, choices were in time as long as they were made in the stimulus window of 1.5 s. Positive feedback 'Outcome: +100' and 'Reward: +100' were shown in green letters, negative feedback ('Outcome: 0', 'Reward: 0', and 'Too slow!") were shown in red letters.

simultaneously. We compared model fits informally using posterior predictive distributions—calculating the same summary statistics on data generated from the fitted model as we did for the empirical data—and formally using the Bayesian Predictive Information Criterion (BPIC; *Ando, 2007*). The former method allows us to assess the absolute quality of fit (*Palminteri et al., 2017*) and detect misfits; the latter provides a model-selection criterion that trades off quality of fit with model complexity (lower BPICs are preferred), ensuring that a better fit is not only due to greater model flexibility.

We first examine results aggregated over difficulty conditions. The posterior predictives of all four RL-EAMs are shown in *Figure 3*, with the top row showing accuracies, and the middle and bottom rows correct and error RT distributions (parameter estimates for all models can be found in

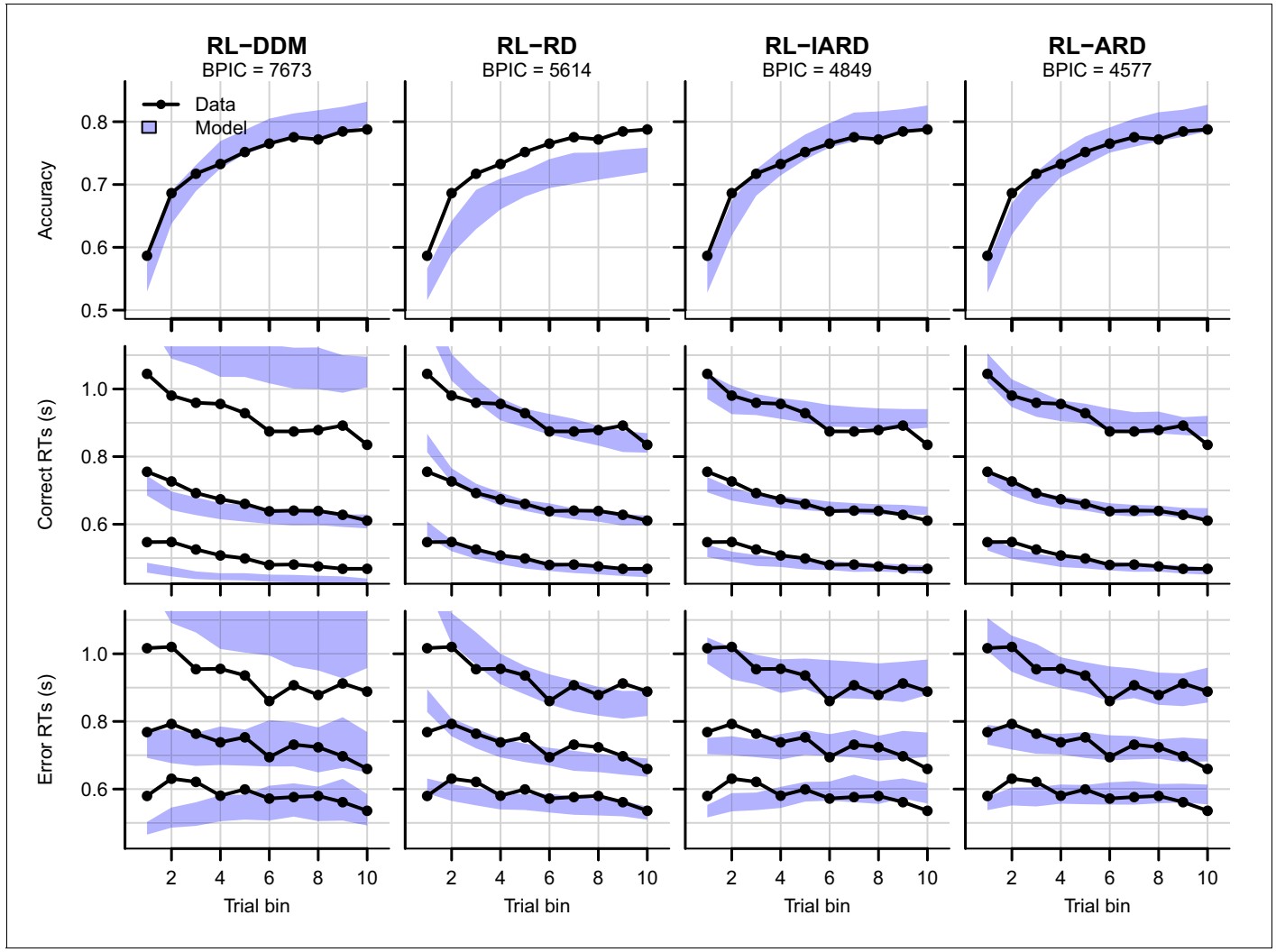

**Figure 3.** Comparison of posterior predictive distributions of the four RL-EAMs. Data (black) and posterior predictive distribution (blue) of the RL-DDM (left column), RL-RD, RL-IARD, and RL-ARD (right column). Top row depicts accuracy over trial bins. Middle and bottom row show 10th, 50th, and 90th RT percentiles for the correct (middle row) and error (bottom row) response over trial bins. Shaded areas correspond to the 95% credible interval of the posterior predictive distributions. All data are collapsed across participants and difficulty conditions.

The online version of this article includes the following figure supplement(s) for figure 3:

**Figure supplement 1.** Comparison of posterior predictive distributions of four additional RL-DDMs.

**Figure supplement 2.** Parameter recovery of the RL-ARD model, using the experimental paradigm of experiment 1.

**Figure supplement 3.** Confusion matrices showing model separability.

**Figure supplement 4.** Empirical (black) and posterior predictive (blue) defective probability densities of the RT distributions estimated using kernel density approximation.

**Table 1.** Posterior parameter estimates (across-subject mean and SD of the median of the posterior distributions) for all models and experiments.

For models including $s_{t0}$, the non-decision time is assumed to be uniformly distributed with bounds $[t0, \ t0 + s_{t0}]$.

**Experiment 1**

| RL-DDM | $\alpha$ | $a$ | $t0$ | $w$ | | | | | BPIC |
|---|---|---|---|---|---|---|---|---|---|
| | 0.14 (0.11) | 1.48 (0.19) | 0.30 (0.06) | 3.21 (1.11) | | | | | 7673 |
| RL-RD | $\alpha$ | $a$ | $t0$ | $V_0$ | $w$ | | | | |
| | 0.12 (0.08) | 2.16 (0.27) | 0.10 (0.04) | 1.92 (0.42) | 3.09 (1.32) | | | | 5613 |
| RL-IARD | $\alpha$ | $a$ | $t0$ | $V_0$ | $w_d$ | | | | |
| | 0.13 (0.12) | 2.05 (0.24) | 0.12 (0.05) | 2.48 (0.43) | 2.36 (0.95) | | | | 4849 |
| RL-ARD | $\alpha$ | $a$ | $t0$ | $V_0$ | $w_d$ | $w_s$ | | | |
| | 0.13 (0.11) | 2.14 (0.26) | 0.11 (0.04) | 2.46 (0.59) | 2.25 (0.78) | 0.36 (0.79) | | | 4577 |
| RL-DDM A1 | $\alpha$ | $a$ | $t0$ | $w$ | $v_{max}$ | | | | |
| | 0.14 (0.12) | 1.49 (0.20) | 0.30 (0.06) | 3.01 (0.66) | 2.81 (0.72) | | | | 7717 |
| RL-DDM A2 | $\alpha$ | $a$ | $t0$ | $w$ | $s_z$ | $s_v$ | | | |
| | 0.14 (0.11) | 1.48 (0.19) | 0.30 (0.06) | 3.21 (1.12) | $1.79e^{-3}$ ($0.4e^{-3}$) | $1.8e^{-3}$ ($0.4e^{-3}$) | | | 7637 |
| RL-DDM A3 | $\alpha$ | $a$ | $t0$ | $w$ | $s_z$ | $s_v$ | $s_{t0}$ | | |
| | 0.13 (0.12) | 1.13 (0.19) | 0.27 (0.06) | 5.31 (2.04) | 0.00 (0.00) | 0.31 (0.13) | 0.37 (0.13) | | 4844 |
| RL-DDM A4 | $\alpha$ | $a$ | $t0$ | $w$ | $v_{max}$ | $s_v$ | $s_z$ | $s_{t0}$ | |
| | 0.13 (0.12) | 1.15 (0.17) | 0.27 (0.06) | 2.02 (0) | 5.16 (1.18) | 0.55 (0.24) | $1.57e^{-3}$ (0) | 0.36 (0.13) | 4884 |
| RL-ALBA | $\alpha$ | $a$ | $t0$ | $V_0$ | $w_d$ | $w_s$ | A | | |
| | 0.13 (0.11) | 3.53 (0.53) | 0.03 (0.00) | 3.03 (0.57) | 2.03 (0.59) | 0.33 (0.78) | 1.73 (0.43) | | 4836 |

Experiment 2

| RL-DDM 1 | $\alpha$ | $a_{spd}/a_{acc}$ | $t0$ | $w$ | | | | |
|---|---|---|---|---|---|---|---|---|
| | 0.13 (0.06) | 1.11 (0.18)/1.42 (0.23) | 0.26 (0.06) | 3.28 (0.66) | | | | 979 |
| RL-DDM 2 | $\alpha$ | $a$ | $t0$ | $w_{spd}/w_{acc}$ | | | | |
| | 0.13 (0.05) | 3.01 (0.63) | 0.26 (0.06) | 3.46 (0.79)/3.01 (0.63) | | | | 1518 |
| RL-DDM 3 | $\alpha$ | $a_{spd}/a_{acc}$ | $t0$ | $w_{spd}/w_{acc}$ | | | | |
| | 0.13 (0.06) | 1.10 (0.18)/1.44 (0.23) | 0.26 (0.06) | 3.11 (0.68)/3.48 (0.72) | | | | 999 |
| RL-ARD 1 | $\alpha$ | $a_{spd}/a_{acc}$ | $t0$ | $V_0$ | $w_d$ | $w_s$ | | |
| | 0.12 (0.05) | 1.45 (0.35)/1.82 (0.35) | 0.15 (0.07) | 2.59 (0.50) | 2.24 (0.53) | 0.47 (0.34) | | −1044 |
| RL-ARD 2 | $\alpha$ | $a$ | $t0$ | $V_0$ | $w_d$ | $w_s$ | $m_{v,spd}$ | |
| | 0.12 (0.05) | 1.83 (0.36) | 0.12 (0.07) | 2.52 (0.53) | 1.83 (0.56) | 0.32 (0.26) | 1.31 (0.20) | −827 |
| RL-ARD 3 | $\alpha$ | $a$ | $t0$ | $V_{0,spd}/V_{0,acc}$ | $w_d$ | $w_s$ | | |
| | 0.12 (0.05) | 1.83 (0.35) | 0.12 (0.07) | 3.37 (0.84)/3.37 (0.54) | 2.11 (0.52) | 0.39 (0.30) | | −934 |
| RL-ARD 4 | $\alpha$ | $a_{spd}/a_{acc}$ | $t0$ | $V_0$ | $w_d$ | $w_s$ | $m_{v,spd}$ | |
| | 0.12 (0.05) | 1.04 (0.14)/1.82 (0.35) | 0.15 (0.07) | 2.59 (0.52) | 2.21 (0.51) | 0.44 (0.38) | 1.04 (0.14) | −1055 |
| RL-ARD 5 | $\alpha$ | $a_{spd}/a_{acc}$ | $t0$ | $V_{0,spd}/V_{0,acc}$ | $w_d$ | $w_s$ | | |
| | 0.12 (0.05) | 1.59 (0.40)/1.83 (0.32) | 0.14 (0.06) | 2.92 (0.65)/2.52 (0.50) | 2.21 (0.50) | 0.43 (0.33) | | −1071 |
| RL-ARD 6 | $\alpha$ | $a$ | $t0$ | $V_{0,spd}/V_{0,acc}$ | $w_d$ | $w_s$ | $m_{v,spd}$ | |
| | 0.12 (0.05) | 1.86 (0.35) | 0.12 (0.07) | 4.13 (0.98)/2.40 (0.54) | 2.28 (0.53) | 0.44 (0.33) | 0.84 (0.03) | −897 |
| RL-ARD 7 | $\alpha$ | $a_{spd}/a_{acc}$ | $t0$ | $V_{0,spd}/V_{0,acc}$ | $w_d$ | $w_s$ | $m_{v,spd}$ | |
| | 0.12 (0.05) | 1.61 (0.40)/1.87 (0.32) | 0.14 (0.06) | 3.66 (0.74)/2.52 (0.50) | 2.41 (0.53) | 0.48 (0.38) | 0.82 (0.08) | −1060 |
| RL-DDM A3 1 | $\alpha$ | $a_{spd}/a_{acc}$ | $t0$ | $w$ | $s_z$ | $s_v$ | $s_{t0}$ | |
| | 0.12 (0.05) | 0.81 (0.16)/1.14 (0.17) | 0.23 (0.06) | 4.46 (0.79) | 0.10 (0.01) | 0.18 (0.05) | 0.26 (0.09) | −862 |
| RL-DDM A3 2 | $\alpha$ | $a$ | $t0$ | $w_{spd}/w_{acc}$ | $s_z$ | $s_v$ | $s_{t0}$ | |

Table 1 continued

**Experiment 1**

|  | α | $a_{spd}/a_{acc}$ | t0 | $w_{spd}/w_{acc}$ | $s_z$ | $s_v$ | $s_{t0}$ |  |
|---|---|---|---|---|---|---|---|---|
|  | 0.12 (0.05) | 1.03 (0.14) | 0.24 (0.06) | 18.4 (23.34)/4.44 (0.84) | 0.26 (0.07) | 0.61 (0.50) | 0.28 (0.10) | −325 |
| RL-DDM A3 3 | α | $a_{spd}/a_{acc}$ | t0 | $w_{spd}/w_{acc}$ | $s_z$ | $s_v$ | $s_{t0}$ |  |
|  | 0.12 (0.05) | 0.81 (0.16)/1.14 (0.17) | 0.23 (0.06) | 4.45 (0.83)/4.45 (0.83) | 0.07 (0.00) | 0.17 (0.04) | 0.26 (0.09) | −849 |

**Experiment 3**

| Soft-max | α | β |  |  |  |  |  |  |
|---|---|---|---|---|---|---|---|---|
|  | 0.40 (0.14) | 2.82 (1.1) |  |  |  |  |  | 23,727 |
| RL-DDM | α | a | t0 |  |  |  |  |  |
|  | 0.38 (0.14) | 1.37 (0.24) | 0.24 (0.07) |  |  |  |  | 15,599 |
| RL-ARD | α | a | t0 | $V_0$ | $w_d$ | $w_s$ |  |  |
|  | 0.35 (0.15) | 1.48 (0.34) | 0.13 (0.08) | 1.86 (0.51) | 1.52 (0.63) | 0.23 (0.25) |  | 11,548 |
| RL-DDM A3 | α | a | t0 | w | $s_z$ | $s_v$ | $s_{t0}$ |  |
|  | 0.38 (0.14) | 1.15 (0.22) | 0.22 (0.07) | 2.72 (1.16) | 0.21 (0.09) | 0.28 (0.15) | 0.27 (0.17) | 11,659 |

**Experiment 4**

| RL-ARD (Win-All) | α | a | t0 | $V_0$ | $w_d$ | $w_s$ |  |
|---|---|---|---|---|---|---|---|
|  | 0.10 (0.04) | 1.6 (0.33) | 0.07 (0.05) | 1.14 (0.22) | 1.6 (0.36) | 0.15 (0.26) | 36,512 |

*Table 1*). The RL-DDM generally explains the learning-related increase in accuracy well, and if only the central tendency were relevant it might be considered to provide an adequate account of RT, although correct median RT is systematically under-estimated. However, RT variability and skew are severely over-estimated. The RL-RD largely overcomes the RT distribution misfit, but it overestimates RTs in the first trial bins, and while capturing an increase in accuracy over trials, it is systematically underestimated. The RL-ARD models provide the best explanation of all key aspects of the data: except for a slight underestimation of accuracy in early trial bins (largely shared with the RL-DDM), they capture accuracy well, and like the RL-RD, they capture the RT distributions well, but without overpredicting the RTs in the early trials. The two RL-ARD models do not differ greatly in fit, except that the limited version slightly underestimates the decrease in RT with learning.

*Figure 4* shows the data and RL-ARD model fit separated by difficulty (see *Figure 4—figure supplement 1* for equivalent RL-DDM fits, which again fail to capture RT distributions). The RL-ARD model displays the same excellent fit as to data aggregated over difficulty, except that it underestimates accuracy in early trials in the easiest condition (*Figure 4*, bottom right panel). Further inspections of the data revealed that 17 participants (31%) reached perfect accuracy in the first bin in this condition. Likely, they guessed correctly on the first occurrence of the easiest choice pair, repeated their choice, and received too little negative feedback in the next repetitions to change their choice strategy. *Figure 4—figure supplement 2* shows that, with these 17 participants removed, the overestimation is largely mitigated. The delta rule assumes learning from feedback, and so cannot explain such high early accuracies. Working memory processes could have aided performance in the easiest condition, since the total number of stimuli pairs was limited and feedback was quite reliable, making it relatively easy to remember correct-choice options (*Collins and Frank, 2018*; *Collins and Frank, 2012*; *McDougle and Collins, 2020*).

## Reward magnitude and Q-value evolution

Q-values represent the participants' internal beliefs about how rewarding each choice option is. The RL-lARD and RL-DDM assume drift rates are driven only by the difference in Q-values (*Figure 5*), and both underestimate the learning-related decrease in RTs. Similar RL-DDM underestimation has been detected before (*Pedersen et al., 2017*), with the proposed remedy being a decrease in the decision bound with time (but with no account of RT distributions). The RL-ARD explains the additional speed-up through the increasing *sum* of Q-values over trials (*Figure 5C*), which in turn increases drift rates (*Figure 5D*). In line with observations in perceptual decision-making

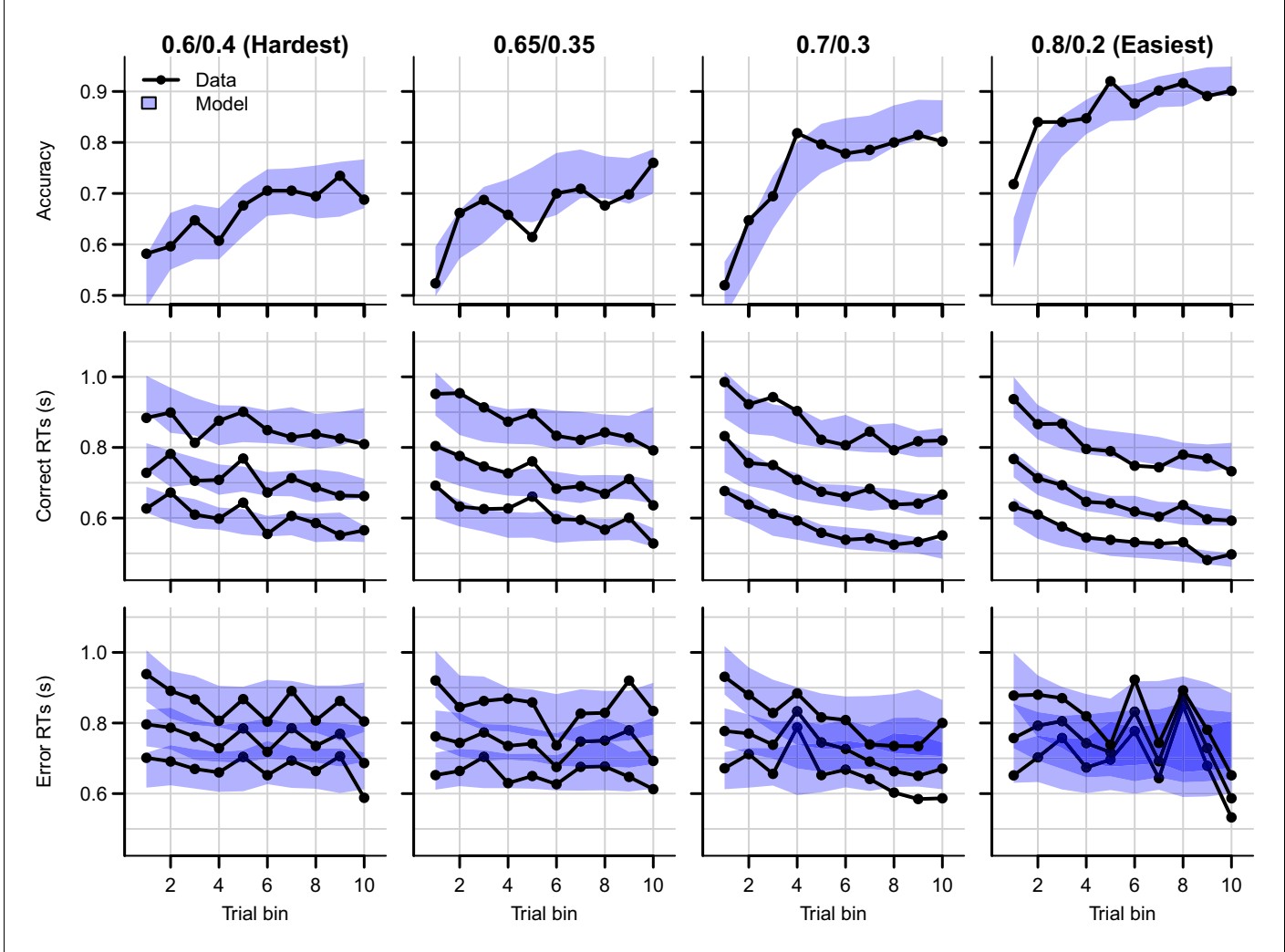

**Figure 4.** Data (black) and posterior predictive distribution of the RL-ARD (blue), separately for each difficulty condition. Column titles indicate the reward probabilities, with 0.6/0.4 being the most difficult, and 0.8/0.2 the easiest condition. Top row depicts accuracy over trial bins. Middle and bottom rows show 10th, 50th, and 90th RT percentiles for the correct (middle row) and error (bottom row) response over trial bins. Shaded areas correspond to the 95% credible interval of the posterior predictive distributions. All data and fits are collapsed across participants.

The online version of this article includes the following figure supplement(s) for figure 4:

**Figure supplement 1.** Data (black) and posterior predictive distribution of the RL-DDM (blue), separately for each difficulty condition.

**Figure supplement 2.** Data (black) and posterior predictive distribution of the RL-ARD (blue), separately for each difficulty condition, excluding 17 subjects which had perfect accuracy in the first bin of the easiest condition.

**Figure supplement 3.** Posterior predictive distribution of the RL-ALBA model on the data of experiment 1, with one column per difficulty condition.

**Figure supplement 4.** Data (black) and posterior predictive distribution of the RL-ARD (blue), separately for each difficulty condition.

(*van Ravenzwaaij et al., 2020*), the effect of the expected reward magnitude on drift rate is smaller (on average, $w_s = 0.36$) than that of the Q-value difference ($w_d = 2.25$) and the urgency signal ($V_0 = 2.45$). Earlier work using an RL-DDM (*Fontanesi et al., 2019a*) showed that higher reward magnitudes decrease RTs in reinforcement learning paradigms. There, the reward magnitude effect on RT was accounted for by allowing the threshold to change as a function of magnitude. However, this requires participants to rapidly adjust their threshold based on the identity of the stimuli, something that is usually not considered possible in EAMs (*Donkin et al., 2011*; *Ratcliff, 1978*). The RL-ARD avoids this problem, with magnitude effects entirely mediated by drift rates, and our results show that the expected reward magnitudes influence RTs due to learning even in the absence of a reward

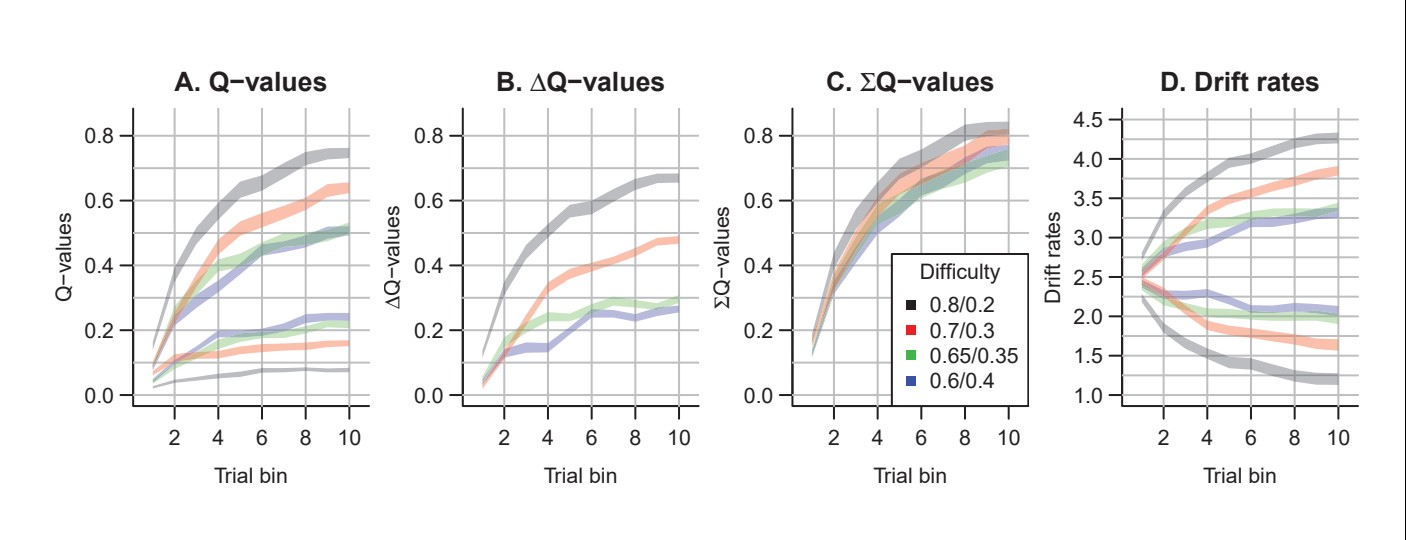

**Figure 5.** The evolution of Q-values and their effect on drift rates in the RL-ARD. A depicts raw Q-values, separate for each difficulty condition (colors). B and C depict the Q-value differences and the Q-value sums over time. The drift rates (D) are a weighted sum of the Q-value differences and Q-value sums, plus an intercept.

magnitude manipulation. Because the sum affects each accumulator equally, it changes RT with little effect on accuracy.

## Speed-accuracy trade-off

Speed-accuracy trade-off (SAT) refers to the ability to strategically trade-off decision speed for decision accuracy (*Bogacz et al., 2010*; *Pachella and Pew, 1968*; *Ratcliff and Rouder, 1998*). As participants can voluntarily trade speed for accuracy, RT and accuracy are not independent variables, so analysis methods considering only one of these variables while ignoring the other can be misleading. EAMs simultaneously consider RTs and accuracy and allow for estimation of SAT settings. The classical explanation in the DDM framework (*Ratcliff and Rouder, 1998*) holds that participants adjust their SAT by changing the decision threshold: increasing thresholds require a participant to accumulate more evidence, leading to slower but more accurate responses.

Empirical work draws a more complex picture. Several papers suggest that in addition to thresholds, drift rates (*Arnold et al., 2015*; *Heathcote and Love, 2012*; *Ho et al., 2012*; *Rae et al., 2014*; *Sewell and Stallman, 2020*) and sometimes even non-decision times (*Arnold et al., 2015*; *Voss et al., 2004*) can be affected. Increases in drift rates in a race model could indicate an urgency signal, implemented by drift gain modulation, with qualitatively similar effects to collapsing thresholds over the course of a decision (*Cisek et al., 2009*; *Hawkins et al., 2015*; *Miletić, 2016*; *Miletić and van Maanen, 2019*; *Murphy et al., 2016*; *Thura and Cisek, 2016*; *Trueblood et al., 2021*; *van Maanen et al., 2019*). In cognitively demanding tasks, it has been shown that two distinct components of evidence accumulation (quality and quantity of evidence) are affected by SAT manipulations, with quantity of evidence being analogous to an urgency signal (*Boag et al., 2019b*; *Boag et al., 2019a*). Recent evidence suggests that different SAT manipulations can affect different psychological processes: cue-based manipulations that instruct participants to be fast or accurate lead to overall threshold adjustments, whereas deadline-based manipulations can lead to a collapse of thresholds (*Katsimpokis et al., 2020*).

Here, we apply an SAT manipulation to an instrumental learning task (*Figure 2C*). The paradigm differs from experiment one by the inclusion of a cue-based instruction to either stress response *speed* ('SPD') or response *accuracy* ('ACC') prior to each choice (randomly interleaved). Furthermore, on speed trials, participants had to respond within 0.6 s to receive a reward. Feedback was determined based on both the choice's probabilistic outcome ('+100' or '+0') and the RT: On trials where participants responded too late, they were additionally informed of the reward associated with their choice, had they been in time, so that they always received the feedback required to learn from their

choices. After exclusions (see Materials and methods), data from 19 participants (324 trials each) were included in the analyses.

We used two mixed effects models to confirm the effect of the manipulation. A linear model predicting RT confirmed an interaction between trial bin and cue (b = 0.014, SE = $1.53*10^{-3}$, 95% CI [0.011, 0.017], $p < 10^{-16}$), a main effect of cue (b = -0.189, SE = $9.5*10^{-3}$, 95% CI [-0.207, -0.170], $p < 10^{-16}$) and a main effect of trial bin (b = -0.015, SE = $1.08*10^{-3}$, 95% CI [-0.018, -0.013], $p < 10^{-16}$). Thus, RTs decreased with trial bin, were faster for the speed condition, but the effect of the cue was smaller for later trial bins. A logistic mixed effects model of choice accuracy showed a main effect of the cue (b = -0.39, SE = 0.13, 95% CI [-0.65, -0.13], $p = 0.003$) and trial bin (b = 0.42, SE = 0.06, 95% CI [0.30, 0.53], $p = 3.1*10^{-12}$), but not for an interaction (b = 0.115, SE = 0.08, 95% CI [-0.05, 0.28], $p = 0.165$). Hence, participants were more often correct in the accuracy condition, and their accuracy increased over trial bins, but there was no evidence for a difference in the increase (on a logit scale) between SAT conditions.

Next, we compared the RL-DDM and RL-ARD. In light of the multiple psychological mechanisms potentially affected by the SAT manipulation, we allowed different combinations of threshold, drift rate, and for the RL-ARD urgency, to vary with the SAT manipulation. We fit three RL-DDM models, varying either threshold, the Q-value weighting on the drift rates parameter (*Sewell and Stallman, 2020*), or both. For the RL-ARD, we fit all seven possible models with different combinations of the threshold, urgency, and drift rate parameters free to vary between SAT conditions.

Formal model comparison (see *Table 1* for all BPIC values) indicates that the RL-ARD model combining response caution and urgency effects provides the best explanation of the data, in line with earlier research in non-learning contexts (*Katsimpokis et al., 2020*; *Miletić and van Maanen, 2019*; *Rae et al., 2014*; *Thura and Cisek, 2016*). The advantage for the RL-ARD was substantial; the best RL-DDM (with only a threshold effect) performed worse than the worst RL-ARD model. The data and posterior predictive distributions of the best RL-DDM model and the winning RL-ARD model are shown in *Figure 6*. As in experiment 1, the RL-DDM failed to capture the shape of RT distributions, although it fit the SAT effect on accuracy and median RTs. The RL-ARD model provides a much better account of the RT distributions, including the differences between SAT conditions. In *Figure 6—figure supplement 1*, we show that adding non-decision time variability to the RL-DDM mitigates some of the misfit of the RT distributions, although it still consistently under-predicted the 10th percentile in the accuracy condition. Further, this model was still substantially outperformed by the RL-ARD in formal model selection (ΔBPIC = 209), and non-decision time variability was estimated as much greater than what is found in non-learning contexts, raising the question of its psychological plausibility.

Both RL-DDM and RL-ARD models tended to underestimate RTs and choice accuracy in the early trial bins in the accuracy emphasis condition. As in experiment 1, working memory may have contributed to the accurate but slow responses in the first trial bin for the accuracy condition (*Collins and Frank, 2018*; *Collins and Frank, 2012*; *McDougle and Collins, 2020*).

## Reversal learning

Next, we tested whether the RL-ARD can capture changes in accuracy and RTs caused by a perturbation in the learning process due to reversals in reward contingencies. In the reversal learning paradigm (*Behrens et al., 2007*; *Costa et al., 2015*; *Izquierdo et al., 2017*) participants first learn a contingency between choice options and probabilistic rewards (the acquisition phase) that is then suddenly reversed without any warning (the reversal phase). If the link between Q-values and decision mechanisms as proposed by the RL-ARD underlies decisions, the model should be able to account for the behavioral consequences (RT distributions and decisions) of Q-value changes induced by the reversal.

Our reversal learning task had the same general structure as experiment 1 (*Figure 1*), except for the presence of reversals. Forty-seven participants completed four blocks of 128 trials each. Within each block, two pairs of stimuli were randomly interleaved. Between trials 61 and 68 (uniformly sampled) in each block, the reward probability switched between stimuli, such that stimuli that were correct during acquisition were incorrect after reversal (and vice versa). Participants were not informed of the reversals prior to the experiment, but many reported noticing them.

Data and the posterior predictive distributions of the RL-DDM and the RL-ARD models are shown in *Figure 7*. Both models captured the change in choice proportions after the reversal reasonably

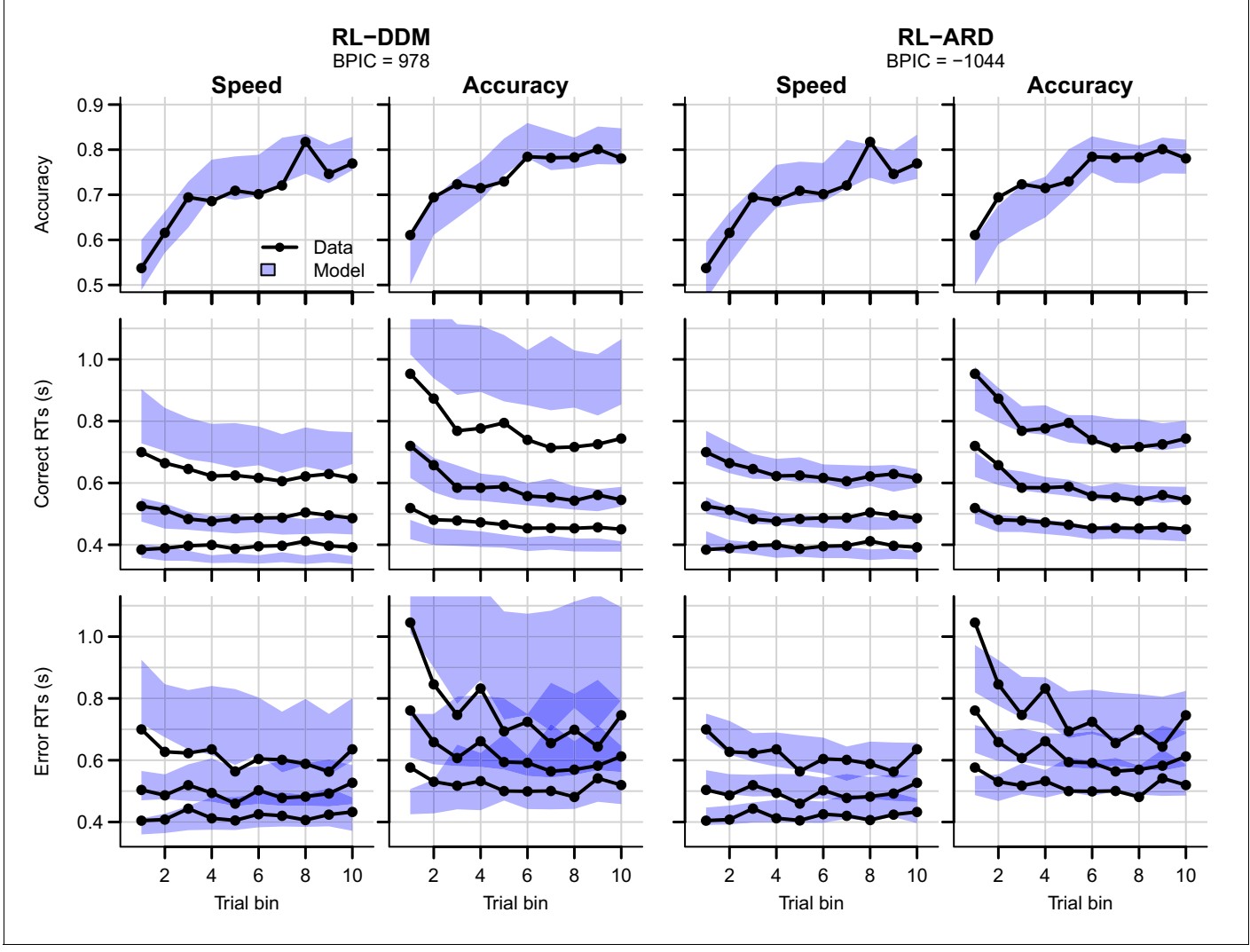

**Figure 6.** Data (black) and posterior predictive distributions (blue) of the best-fitting RL-DDM (left columns) and the winning RL-ARD model (right columns), separate for the speed and accuracy emphasis conditions. Top row depicts accuracy over trial bins. Middle and bottom row show 10th, 50th, and 90th RT percentiles for the correct (middle row) and error (bottom row) response over trial bins. Shaded areas in the middle and right column correspond to the 95% credible interval of the posterior predictive distribution.

The online version of this article includes the following figure supplement(s) for figure 6:

**Figure supplement 1.** Data (black) of experiment 2 and posterior predictive distribution (blue) of the RL-DDM A3 with separate thresholds for the SAT conditions, and between-trial variabilities in drift rates, start points, and non-decision times.

**Figure supplement 2.** Parameter recovery of the RL-ARD model, using the experimental paradigm of experiment 2.

**Figure supplement 3.** Mean RT (left column) and choice accuracy (right column) across trial bins (x-axis) for experiments 2 and 3 (rows).

**Figure supplement 4.** Empirical (black) and posterior predictive (blue) defective probability densities of the RT distributions of experiment 2, estimated using kernel density approximation.

**Figure supplement 5.** Data (black) and posterior predictive distributions (blue) of the best-fitting RL-DDM (left columns) and the winning RL-ARD model (right columns), separate for the speed and accuracy emphasis conditions.

well, although they underestimate the speed of change. In *Figure 7—figure supplement 1*, we show that the same is true for a standard soft-max model, suggesting that the learning rule is the cause of this problem. Recent evidence indicates that, instead of only estimating expected values of both choice options by error-driven learning, participants may additionally learn the task structure, estimate the probability of a reversal occurring and adjust choice behavior accordingly. Such a model-based learning strategy could increase the speed with which choice behavior changes after a

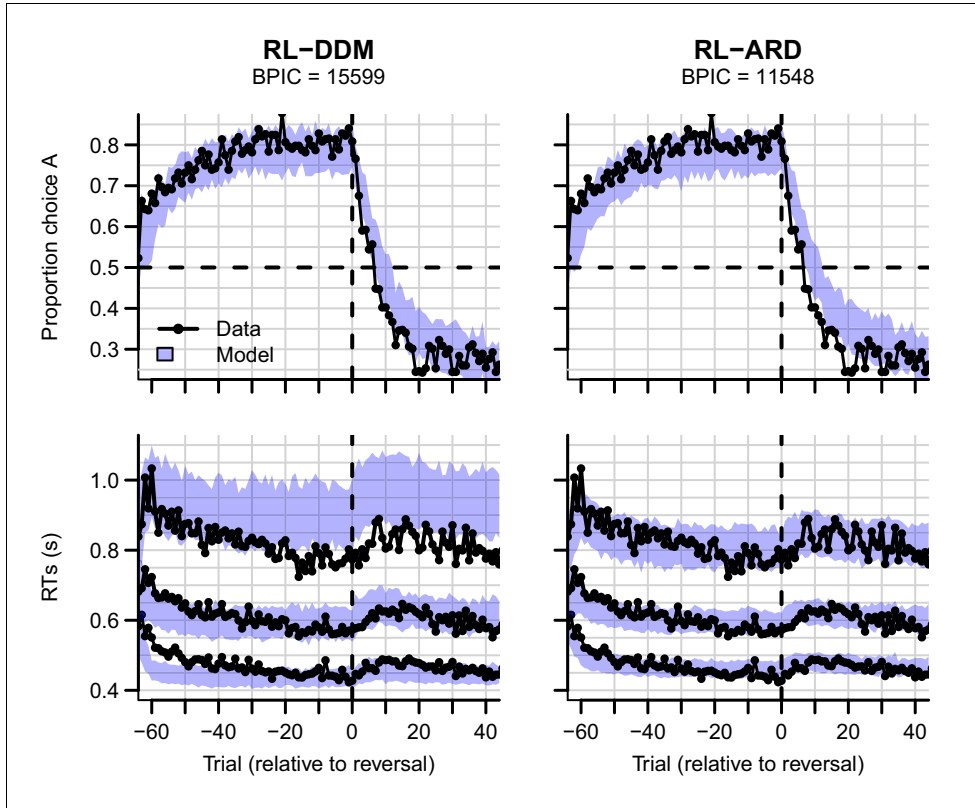

**Figure 7.** Experiment three data (black) and posterior predictive distributions (blue) for the RL-DDM (left) and RL-ARD (right). Top row: choice proportions over trials, with choice option A defined as the high-probability choice before the reversal in reward contingencies. Bottom row: 10th, 50th, and 90th RT percentiles. The data are ordered relative to the trial at which the reversal first occurred (trial 0, with negative trial numbers indicated trials prior to the reversal). Shaded areas correspond to the 95% credible interval of the posterior predictive distributions.

The online version of this article includes the following figure supplement(s) for figure 7:

**Figure supplement 1.** Data (black) of experiment 3 and posterior predictive of a standard soft-max learning model (blue).

**Figure supplement 2.** Data (black) of experiment 3 and posterior predictive distribution (blue) of the RL-DDM A3 (with between-trial variabilities in drift rates, start points, and non-decision times).

**Figure supplement 3.** Parameter recovery of the RL-ARD model, using the experimental paradigm of experiment 3.

**Figure supplement 4.** Empirical (black) and posterior predictive (blue) defective probability densities of the RT distributions of experiment 3, estimated using kernel density approximation.

**Figure supplement 5.** Experiment three data (black) and posterior predictive distributions (blue) for the RL-DDM (left) and RL-ARD (right).

---

reversal (*Costa et al., 2015*; *Izquierdo et al., 2017*; *Jang et al., 2015*), but as yet a learning rule that implements this strategy has not been developed.

The change in RT around the reversal was less marked than the change in choice probability. Once again, the RL-DDM overestimates variability and skew. Both models fit the effects of learning and reversal similarly, but the fastest responses for the RL-DDM decrease much too quickly during initial learning and the reduction in speed for the slowest responses due to the reversal is strongly overestimated. The RL-ARD provides a much better account of the shape of the RT distributions, and furthermore captures the increase in entire RT *distributions* (instead of only the median) after the reversal point. Formal model comparison also very strongly favors the RL-ARD over the RL-DDM ($\Delta BPIC = 4051$). *Figure 7—figure supplement 2* provides model comparisons to RL-DDMs with between-trial variability parameters, which lead to the same conclusion.

A notable aspect of the data is that choice behavior stabilizes approximately 20 trials after the reversal, whereas RTs remain high compared to just prior to the reversal point for up to ~40 trials. The RL-ARD explains this behavior through relatively high Q-values for the choice option that was correct during the acquisition (but not reversal) phase (i.e. choice A). *Figure 8* depicts the evolution of Q-values, Q-value differences and sums, and drift rates in the RL-ARD model. The Q-values for both choice options increase until the reversal (*Figure 8A*), with a much faster increase for $Q_A$. At the reversal $Q_A$ decreases and $Q_B$ increases, but as $Q_A$ decreases faster than $Q_B$ increases there is a temporary decrease in Q-value sums (*Figure 8C*). After approximately 10 trials post-reversal, $Q_B$ is higher than for $Q_A$, which flips the sign of the Q-value differences (*Figure 8B*). However, $Q_A$ *after* the reversal remains higher than the $Q_B$ *before* the reversal, which causes the (absolute) Q-value differences to be lower after the reversal than before. As a consequence, the drift rates for B after the reversal remain lower than the drift rates for A before the reversal, which increases RT. Clearly, it is important to take account of the sum of inputs to accumulators as well as the difference between them in order to provide an accurate account of the effects of learning.

## Multi-alternative choice

Finally, we again drew on the advantage framework (*van Ravenzwaaij et al., 2020*) to extend the RL-ARD to multi-alternative choice tasks, a domain where the RL-DDM cannot be applied. As in the two-choice case, the multi-alternative RL-ARD assumes one accumulator per pairwise difference between choice options. With three choice options (e.g. 1, 2, 3), there are six (directional) pairwise differences (1-2, 1-3, 2-1, 2-3, 3-1, 3-2), and therefore six accumulators (see *Figure 9*). All accumulators are assumed to race toward a common threshold, with their drift rates determined by the advantage framework's combination of an urgency, an advantage, and a sum term. Since each response is associated with two accumulators (e.g. for option 1, one accumulating the advantage 1–2, and the other accumulating the advantage 1–3), a stopping rule is required to determine when a commitment to a response is made and evidence accumulation stops. Following Van Ravenzwaaij et al., we used the Win-All stopping rule, which proposes that the first response option for which *both* accumulators have reached their thresholds wins. RT is the first passage time of the *slowest* of these two winning accumulators, plus non-decision time.

To test how well the Win-All RL-ARD can explain empirical data, we performed a fourth experiment in which participants were required to repeatedly make decisions between three choice

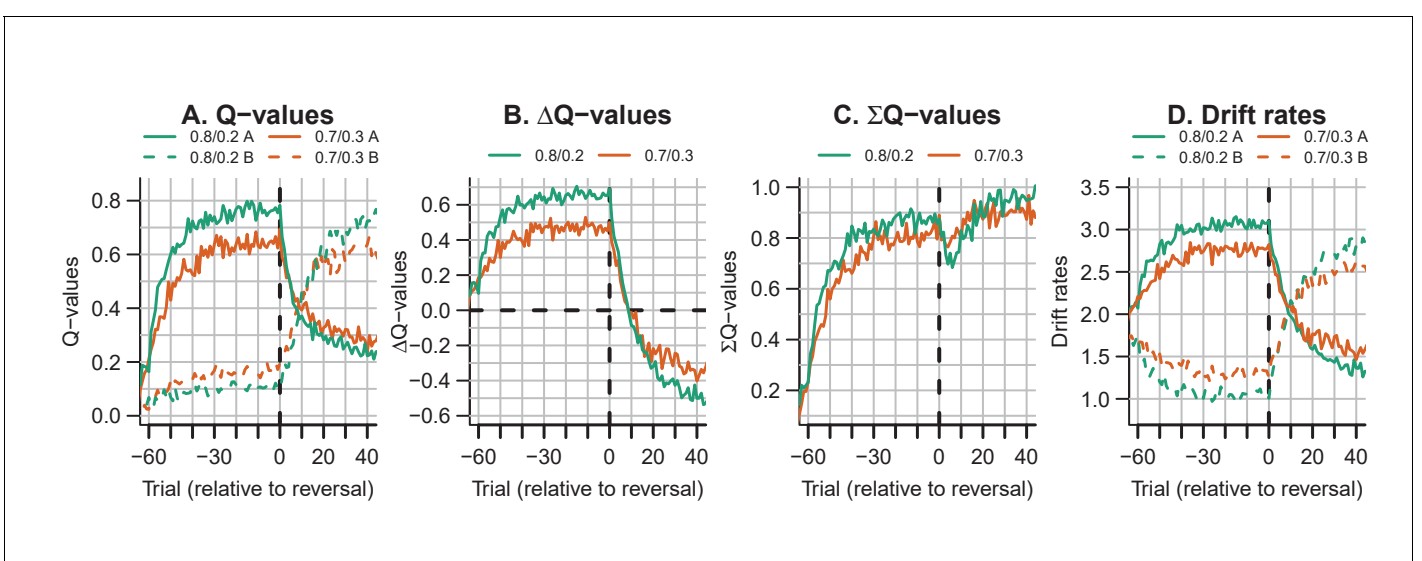

**Figure 8.** The evolution of Q-values and their effect on drift rates in the RL-ARD in experiment 3, aggregated across participants. Left panel depicts raw Q-values, separate for each difficulty condition (colors). The second and third panel depict the Q-value differences and the Q-value sums over time. The drift rates (right panel) are a weighted sum of the Q-value differences and the Q-value sums, plus an intercept. Choice A (solid lines) refers to the option that had the high probability of reward during the acquisition phase, and choice B (dashed lines) to the option that had the high probability of reward after the reversal.

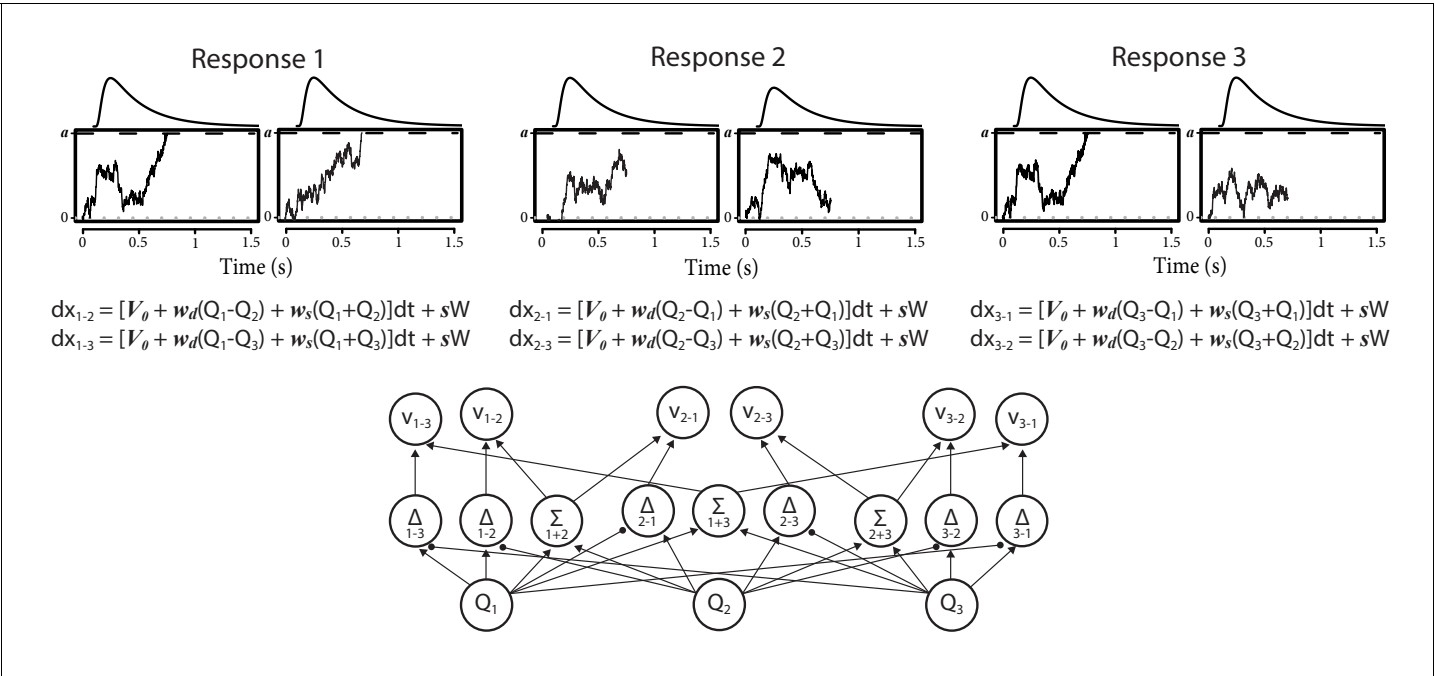

**Figure 9.** Architecture of the three-alternative RL-ARD. In three-choice settings, there are three Q-values. The multi-alternative RL-ARD has one accumulator per directional pairwise difference, hence there are six accumulators. The bottom graph visualizes the connections between Q-values and drift rates ($V_0$ is left out to improve readability). The equations formalize the within-trial dynamics of each accumulator. Top panels illustrate one example trial, in which both accumulators corresponding to response option 1 reached their thresholds. In this example trial, the model chose option 1, with the RT determined by the slowest of the winning accumulators (here, the leftmost accumulator). Decision-related parameters $V_0, w_d, w_s, a, t0$ are all identical across the six accumulators.

options (*Figure 10*). Within each of the four blocks, there were four randomly interleaved stimulus triplets that differed in difficulty (defined as the difference in reward probability between target stimulus and distractors) and reward magnitude (defined as the average reward probability): 0.8/0.25/ 0.25 (easy, high magnitude), 0.7/0.3/0.3 (hard, high magnitude), 0.7/0.15/0.15 (easy, low magnitude), and 0.6/0.2/0.2 (hard, low magnitude). This enabled us to simultaneously test whether the RL-ARD can account for a manipulation of difficulty and mean reward magnitude. Furthermore, we predicted that the requirement to learn 12 individual stimuli (per block) would interfere with the participants' ability to rely on working memory (*Collins and Frank, 2012*), and therefore expected that the RL-ARD would provide a better account of accuracy in the early trial bins compared to experiments 1 and 2. After exclusions (see Materials and methods), data from 34 participants (432 trials each) were included in the analyses.

Data and posterior predictive distributions of the RL-ARD are shown in *Figure 11*. The top row represents accuracy, the middle row the RT quantiles corresponding to the correct (target) choice option, and the bottom row the RT quantiles of the incorrect choices (collapsed across the two distractor response options). Compared to experiments 1–3, the task was substantially more difficult, as evidenced by the relatively low accuracies. The RL-ARD model was able to account for all patterns in the data, including the increase in accuracy and decrease in RTs due to learning, the shape of the full RT distributions, as well as the difficulty and magnitude effects. Furthermore, there was a decrease in variance in the error RTs due to learning (the 10th quantile RTs even mildly *increased*), which was also captured by the model. Note finally that, contrary to experiments 1 and 2, the model did not underestimate the accuracy in early bins in the easy conditions, which suggests that the influence of working memory was, as predicted, more limited than in earlier experiments. A parameter recovery study (*Figure 11—figure supplement 1*) demonstrated that the model's parameters could be recovered accurately.

Notably, *Figure 11* suggests that the effects of the magnitude manipulation were larger in the hard than in the easy condition. As in previous experiments, we inspected the Q-value evolution

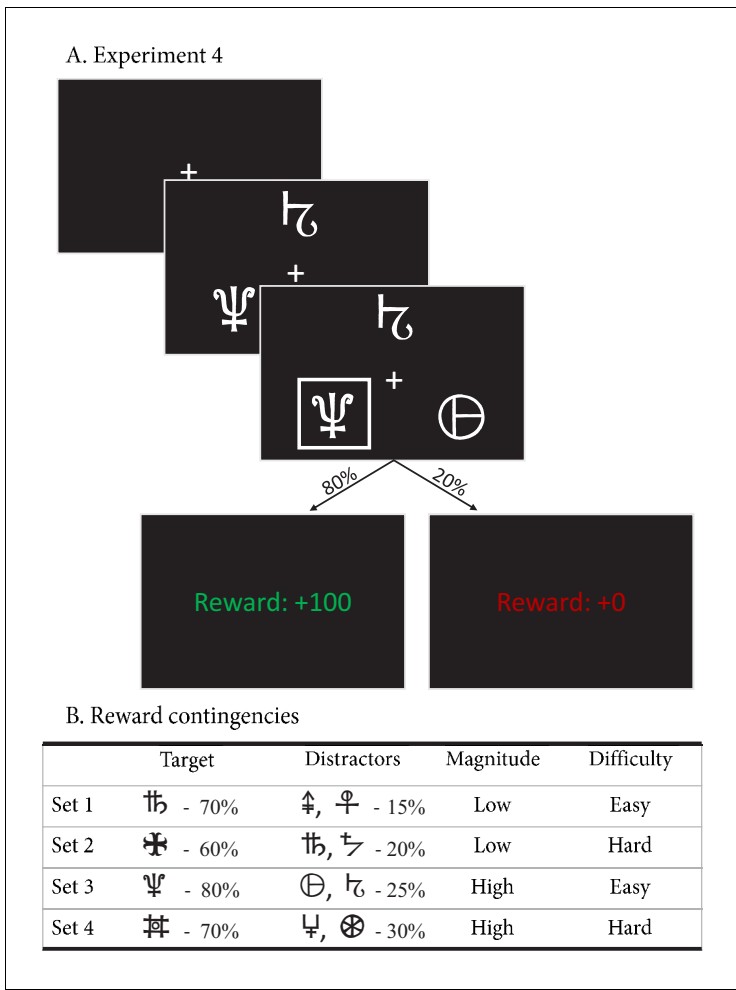

**Figure 10.** Experimental paradigm of experiment 4. (**A**) Example trial of experiment 4. Each trial started with a fixation cross, followed by the stimulus (three choice options; until the subject made a choice, up to 3 s), a brief highlight of the choice, and the choice's reward was shown. (**B**) Reward contingencies for the target stimulus and two distractors per condition. Percentages indicate the probability of receiving +100 points (+0 otherwise). Presented symbols are examples, the actual symbols differed per block and participant (counterbalanced to prevent potential item effects from confounding the learning process).

(*Figure 12*) to understand how this interaction arose. As expected, the high magnitude condition led to higher Q-values (*Figure 12A*) than the low magnitude condition, increasing the Q-value sums (*Figure 12C*). However, there was a second effect of the increased magnitude: even though the true reward probability differences were equal between magnitude conditions, the Q-value differences for the response accumulators ($\Delta Q_{T-D}$; *Figure 12B*) were larger in the high compared to the low magnitude condition, particularly for harder choices. As a consequence, *both* the Q-value sum (weighted by median $w_s = 0.15$), and the smaller changes in the Q-value difference (weighted by median $w_d = 1.6$), increased the drift rates for the response accumulators ($v_{T-D}$; *Figure 12D*), which led to higher accuracy and faster responses.

## Discussion

We compared combinations of different evidence-accumulations models with a simple delta reinforcement learning rule (RL-EAMs). The comparison tested the ability of the RL-EAMs to provide a comprehensive account of behavior in learning contexts, not only in terms of the choices made but also the full distribution of the times to make them (RT). We examined a standard instrumental

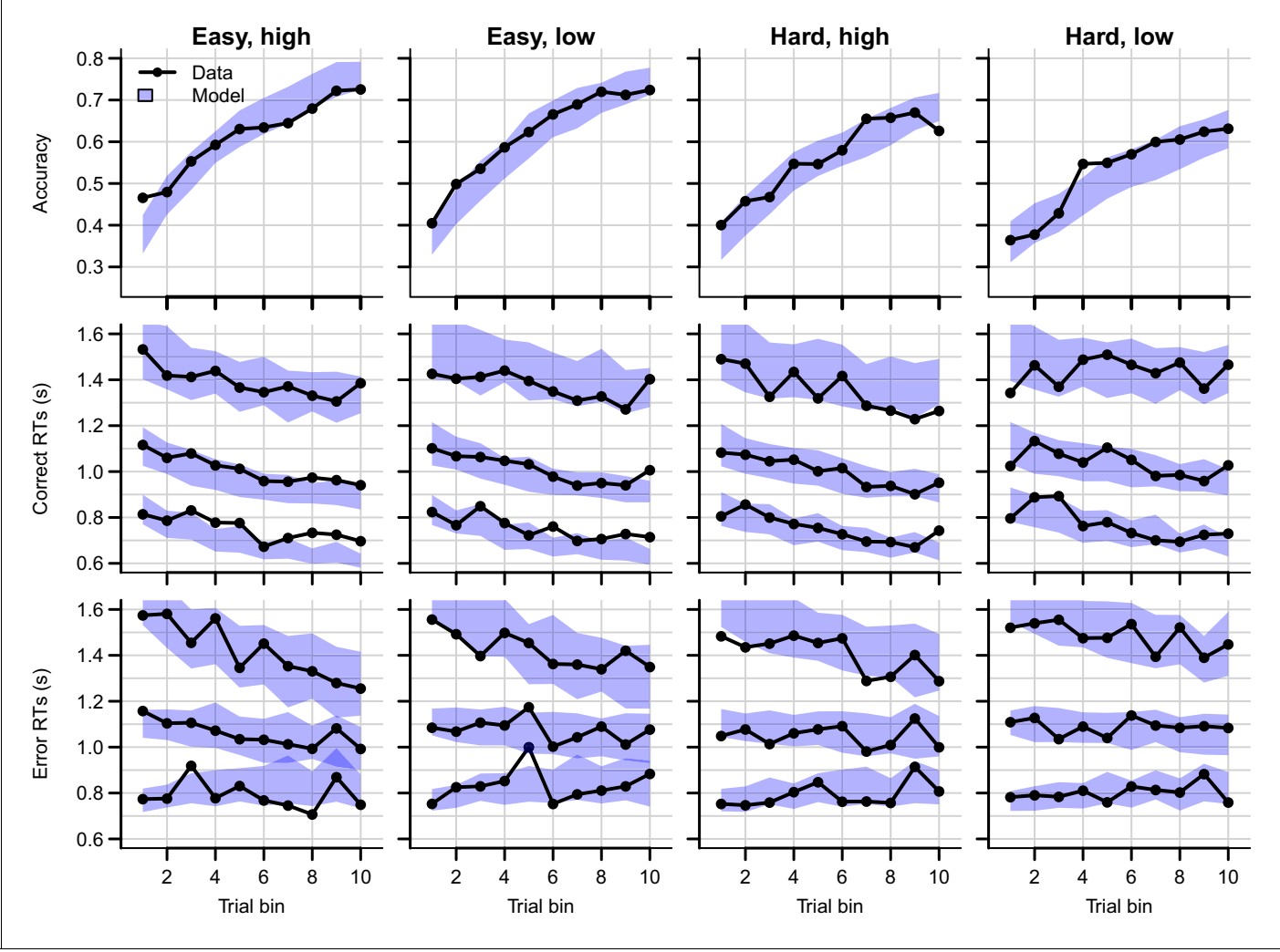

**Figure 11.** Data (black) and posterior predictive distribution of the RL-ARD (blue), separately for each difficulty condition. Column titles indicate the magnitude and difficulty condition. Top row depicts accuracy over trial bins. Middle and bottom rows show 10th, 50th, and 90th RT percentiles for the correct (middle row) and error (bottom row) response over trial bins. The error responses are collapsed across distractors. Shaded areas correspond to the 95% credible interval of the posterior predictive distributions. All data and fits are collapsed across participants.

The online version of this article includes the following figure supplement(s) for figure 11:

**Figure supplement 1.** Parameter recovery of the multi-alterative Win-All RL-ARD model, using the experimental paradigm of experiment 4.

**Figure supplement 2.** Empirical (black) and posterior predictive (blue) defective probability densities of the RT distributions of experiment 4, estimated using kernel density approximation.

**Figure supplement 3.** Data (black) and posterior predictive distribution of the RL-ARD (blue), separately for each difficulty condition of experiment 4.

learning paradigm (*Frank et al., 2004*) that manipulated the difference in rewards between binary options (i.e. decision difficulty). We also examined two elaborations of that paradigm testing key phenomena from the decision-making and learning literatures, speed-accuracy trade-offs (SAT), and reward reversals, respectively. Our benchmark was the dual threshold Diffusion Decision Model (DDM; *Ratcliff, 1978*), which has been used in almost all previous RL-EAM research, but has not been compared to other RL-EAMs, and has not been thoroughly evaluated on its ability to account for RT distributions in learning tasks. Our comparison used several different racing diffusion (RD) models, where decisions depend on the winner of a race between single barrier diffusion processes.

The RL-DDM provided a markedly inferior account to the other models, consistently overestimating RT variability and skew. As these aspects of behavior are considered critical in evaluating models in decision-making literature (*Forstmann et al., 2016*; *Ratcliff and McKoon, 2008*; *Voss et al.,*

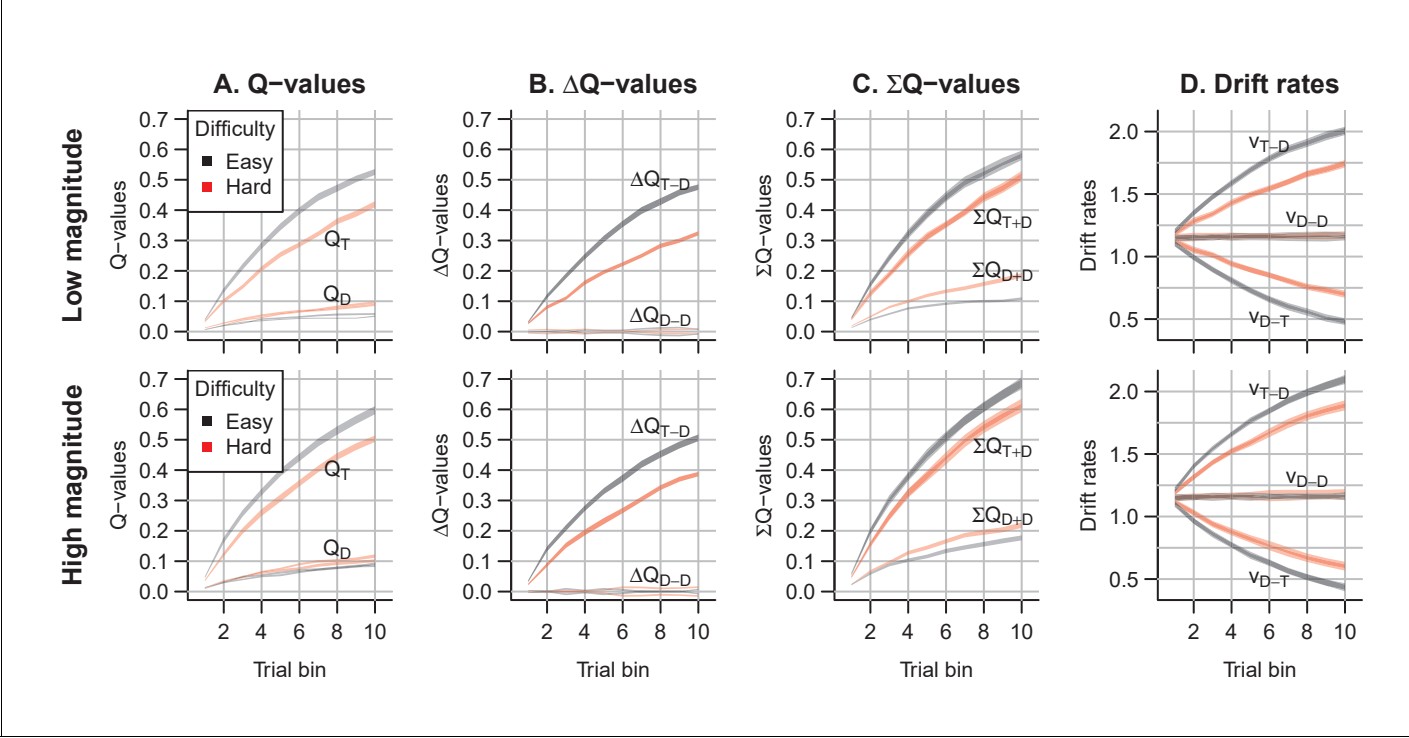

**Figure 12.** Q-value evolution in experiment 4. Top row corresponds to the low magnitude condition, bottom to the high magnitude condition. Colors indicate choice difficulty. (**A**) Q-values for target ($Q_T$) and distractor stimuli ($Q_D$). (**B**) Difference in Q-values, for target – distractor ($\Delta Q_{T-D}$) and between the two distractors ($\Delta Q_{D-D}$). The Q-value difference $\Delta Q_{D-T}$ is omitted from the graph to aid readability (but $\Delta Q_{D-T} = -\Delta Q_{T-D}$). (**C**) Sum of Q-values. (**D**) Resulting drift rates for target response accumulators ($v_{T-D}$), and accumulators for the distractor choice options ($v_{D-T}$, $v_{D-D}$). Note that within each condition, there is a single Q-value trace per choice option, but since there are two distractors, there are two overlapping traces for $\Delta Q_{T-D}$, $\Sigma Q_{T+D}$, and for all drift rates.

2013), our results question whether the RL-DDM provides an adequate model of instrumental learning. Furthermore, the DDM carries with it two important theoretical limitations. First, it can only address binary choice. This is unfortunate given that perhaps the most widely used clinical application of reinforcement learning, the Iowa gambling task (*Bechara et al., 1994*), requires choices among four options. Second, the input to the DDM combines the evidence for each choice (i.e., 'Q' values determined by the learning rule) into a single difference, and so requires extra mechanisms to account for known effects of overall reward magnitude (*Fontanesi et al., 2019a*). Although there are potential ways that the RL-DDM might be modified to account for magnitude effects, such as increasing between-trial drift rate variability in proportion to the mean rate (*Ratcliff et al., 2018*), its inability to extend beyond binary choice remains an enduring impediment.

The best alternative model that we tested, the RL-ARD (advantage racing diffusion), which is based on the recently proposed advantage accumulation framework (*van Ravenzwaaij et al., 2020*), remedied all of these problems. The input to each accumulator is the weighted sum of three components: stimulus independent 'urgency', the difference between evidence for the choice corresponding to the accumulator and the alternative (the advantage), and the sum of the two evidence values. The urgency component had a large effect in all fits and played a key role in explaining the effect of speed-accuracy trade-offs. Although an urgency mechanism such as collapsing bounds (*Boehm et al., 2016*; *Bowman et al., 2012*; *Hawkins et al., 2015*; *Milosavljevic et al., 2010*) or gain modulation (*Boehm et al., 2016*; *Churchland et al., 2008*; *Ditterich, 2006*; *Hawkins et al., 2015*) could potentially improve the fits of the RL-DDM, fitting DDMs with such mechanisms is computationally very expensive, usually requiring the researcher to use simulation-based approximations (e.g. *Turner and Sederberg, 2014*). This expense becomes infeasible in the case of RL-EAMs since these models assume different drift rates per trial, requiring the simulation of an entire dataset *per trial* in the empirical data (for each iteration of the MCMC sampling process). Furthermore, the origin

of the concept of urgency lies in studies using racing accumulator models (*Ditterich, 2006*; *Mazurek et al., 2003*; *Reddi and Carpenter, 2000*), which was only later incorporated in the DDM (*Milosavljevic et al., 2010*); the implementation in the RL-ARD remains conceptually close to the early proposals.

The advantage component of the RL-ARD, which is similar to the input to the DDM, was strongly supported over a model in which each accumulator only receives evidence favoring its own choice. The sum component provides a simple and theoretically transparent way to deal with reward magnitude effects in instrumental learning. Despite having the weakest effect among the three components, the sum was clearly necessary to provide an accurate fit to our data. It also played an important role in explaining the effect of reward reversals.

In all our models, we assumed a linear function to link Q-value differences to drift rates. This may not be adequate in all settings. For example, *Fontanesi et al., 2019a* showed (within an RL-DDM) that a non-linear linking function provided a better fit to their data. This could be caused by a non-linear mapping between objective values and subjective values; the account of perceptual choice using the advantage framework (*van Ravenzwaaij et al., 2020*) relied on a logarithmic mapping between (objective) luminance and (subjective) brightness magnitudes. Prospect Theory (e.g. *Tversky and Kahneman, 1992*) also assumes that increasingly large objective values relative to a reference point lead to increasingly small increases in subjective value. Such non-linear effects only become evident for sufficiently large differences over an appropriate range. Although in our experiments the RL-ARD was able to explain the data well using a simple linear function, future applications may need to explicitly incorporate a non-linear value function.

Finally, we showed that the RL-ARD can be extended to multi-alternative choice. In a three-alternative instrumental learning experiment, it accurately predicted the learning curves and full RT distributions for four conditions that differed in difficulty and reward magnitude. Furthermore, examination of Q-value evolution clarified how the reward magnitude manipulation led to the observed behavioral effects. Notably, the number of choice options only changes the architecture of the RL-ARD while the number of parameters remains constant, and all accumulators in the model remain driven by the same three components: an urgency signal, an advantage, and a sum component. As a consequence, parametric complexity does not increase with number of choices and the model remained fully recoverable despite the relatively low number of trials.

It is perhaps surprising that the RL-DDM consistently overestimated RT variability and skewness given that the DDM typically provides much better fits to data from perceptual decision-making tasks without learning. The inclusion of between-trial variability in non-decision times partially mitigated the misfit but required an implausibly high non-decision time variability, and model comparisons still favored the RL-ARD. Previous work on the RL-DDM did not investigate this issue. In many RL-DDM papers, RT distributions are either not visualized at all, or are plotted using (defective) probability density functions on top of a histogram of RT data, making it hard to detect misfit, particularly with respect to skew due to the slow tail of the distribution. One exception is *Pedersen and Frank, 2020*, whose quantile-based plots show the same pattern that we found here of over-estimated variability and skewness for more difficult choice conditions, despite including between-trial variability in non-decision times. In a non-learning context, it has been shown that the DDM overestimates skewness in high-risk preferential choice data (*Dutilh and Rieskamp, 2016*). Together these results suggest that decision processes in value-based decision in general, and instrumental learning tasks in particular, may be fundamentally different from a two-sided diffusion process, and instead better captured by a race model such as the RL-ARD.

In the current work, we chose to use racing diffusion processes over the more often used LBA models for reasons of parsimony: error-driven learning introduces between-trial variability in accumulation rates, which are explicitly modeled in the RL-EAM framework. As the LBA includes between-trial variability in drift rates as a free parameter, multiple parameters can account for the same variance. Nonetheless, exploratory fits (see *Figure 4—figure supplement 3*) confirmed our expectation that an RL-ALBA (Advantage LBA) model fit the data of experiment one well, although formal model comparisons preferred the RL-ARD. Future work might consider completely replacing one or more sources of between trial variability in the LBA with structured fluctuations due to learning and adaption mechanisms.

The parametrization of the ARD model used in the current paper followed the ALBA model proposed by *van Ravenzwaaij et al., 2020*. This parametrization interprets the influence on drift rates

in terms of advantages and magnitudes. However, as both the weights on Q-value differences and sums ($w_d$ and $w_s$) are freely estimated parameters, the equations that define the drift rates can be rearranged as follows:

$$dx_1 = [V_0 + w_e Q_1 - w_i Q_2]dt + sW$$
$$dx_2 = [V_0 + w_e Q_2 - w_i Q_1]dt + sW$$

where $w_e = w_d + w_s$ in the parametrization of *Equation 4*, and $w_i = w_d - w_s$. This re-parametrization shows that each drift rate is determined by an excitatory influence $w_e$ of the Q-value associated with the accumulator, and an inhibitory influence $w_i$ of the Q-value associated with the other accumulator. *Turner, 2019* proposed that inhibition plays an important role in learning tasks. Although the locus of inhibition is different in the two models, there are clear parallels that bear further investigation.

A limitation of the current work is that we collapsed across blocks in analyzing the data of experiments 2, 3, and 4. However, in more detailed explorations (see *Figure 6—figure supplement 3*) there were indications of second-order changes across blocks. In experiment 2, participants were faster in the first trial bin of the second and third block compared to the first block, suggesting additional practice or adaptation effects at the beginning of the experiment. In experiment 3, participants slowed down, and learned the reversal faster, after the first block. This suggests they learned about the presence of reversals in the first block and applied a different strategy in the later blocks. Although it is known that participants increase their learning rates in volatile environments (*Behrens et al., 2007*), this by itself does not explain a decrease in response speed. Potentially, if participants understood the task structure after the first block, model-based strategies, such as estimating the probability of a reversal having occurred, also slowed down responses. In experiment 4, participants were slower (but equally accurate) in the first block compared to the later blocks, suggesting again additional practice or adaptation effects. Future experiments should investigate the nature of these additional adaptation effects.

Although the account of data provided by the RL-ARD model was generally quite accurate, some elements of misfit suggest the need for further model development. RT and accuracy were underestimated in the initial trials of the easiest condition in experiment 1, in the accuracy emphasis condition in experiment 2, and prior to reversals in experiment 3. Furthermore, the RL-ARD model underestimated the speed with which choice probability changed after reversal of stimulus-response mappings. These misfits point to a limited ability to capture the learning-related changes in behavior. This is to some degree unsurprising, since we used a very simple model of error-driven learning. Future work might explore more sophisticated mechanisms, such as multiple learning rates (*Daw et al., 2002*; *Fontanesi et al., 2019a*; *Gershman, 2015*; *Pedersen et al., 2017*) or different learning rules (*Fontanesi et al., 2019b*; *Fontanesi et al., 2019a*). Furthermore, there is clearly a role for working memory in some reinforcement learning tasks (*Collins and Frank, 2018*; *Collins and Frank, 2012*), likely explaining the accurate but slow responses we observed in the early trial bins for easy conditions in experiments 1–3. The multi-alternative choice experiment, with 12 individual stimuli per block, hampered participants' ability to employ working memory strategies, so even early-trial performance was explained by purely error-driven learning.

In summary, we believe that the ARD decision mechanism provides a firm basis for further explorations of the mutual benefits that arise from the combination of reinforcement learning and evidence-accumulation models, providing constraint that is based on a more comprehensive account of data than has been possible in the past. As it stands, the RL-ARD's parameter recovery properties are good even with relatively low trial numbers, making it a suitable measurement model for simultaneously studying learning and decision-making processes, and inter-individual differences therein. Further, the advantage framework extends to multiple choice while maintaining analytical tractability and addressing key empirical phenomena in that domain, such as Hick's Law and response-competition effects (*van Ravenzwaaij et al., 2020*), enabling future applications to clinical settings, such as in the Iowa gambling task (*Bechara et al., 1994*).

## Materials and methods

### Experiment 1

### Participants

61 participants (mean age 21 years old [SD 2.33], 47 women, 56 right handed) were recruited from the subject pool of the department of Psychology, University of Amsterdam, and participated for course credits. All participants had normal or corrected-to-normal vision and gave written informed consent prior to the experiment onset. They did not participate in the other experiments. The study was approved by the local ethics committee.

### Task

The task was an instrumental probabilistic learning task (*Frank et al., 2004*). On each trial, the subject was presented with two abstract symbols (a 'stimulus pair') representing two choice options (see *Figure 2A* for an example trial). Each choice option had a fixed probability of being rewarded with points when chosen, with one choice option always having a higher probability of being rewarded than the other. The task is to discover, by trial and error, which choice options are most likely to lead to rewards, and thereby to collect as many points as possible.

After a short practice block to get familiar with the task, participants completed one block of 208 trials. Four different pairs of abstract symbols were included, each presented 52 times. Stimulus pairs differed in their associated reward probabilities: 0.8/0.2, 0.7/0.3, 0.65/0.35, and 0.6/0.4. The size of the reward, if obtained, was always the same: '+100' (or '+0' otherwise). Reward probabilities were chosen such that they differed only in the between-choice difference in reward probability, leading to varying choice difficulties while keeping the mean reward magnitude fixed.

Participants were instructed to earn as many points as possible, and to always respond before the deadline of 2 s. Feedback consisted of two parts: an 'outcome' and a 'reward'. The outcome corresponded to the probabilistic outcome of the choice, whereas the reward corresponded to the actual number of earned points. When participants responded before the deadline, the reward was equal to the outcome. If they were too late, the outcome was shown to allow participants to learn from their choice, but the reward they received was set to 0 to encourage responding in time. Participants received a bonus depending on the number of points earned (maximum +0.5 course credits, mean received +0.24). The task was coded in PsychoPy (*Peirce et al., 2019*). After this block, participants performed two more blocks of the same task with different manipulations, which are not of current interest.

### Exclusion

Six participants were excluded from analysis: One reported, after the experiment, not to have understood the task, one reported a technical issue, and four did not reach an above-chance accuracy level as determined by a binomial test (accuracy cut-off 0.55, corresponding to $p < 0.05$). The final sample thus consisted of 55 subjects (14 men, mean age 21 years old [SD 2.39], 51 right-handed).

### Cognitive modeling

The main analysis consists of fitting four RL-EAMs to the data and comparing the quality of the fits penalized by model complexity. We compared four different decision models: the DDM (*Ratcliff, 1978*), a racing diffusion (*Boucher et al., 2007*; *Logan et al., 2014*; *Purcell et al., 2010*; *Turner, 2019*) model, and two Advantage Racing Diffusion (ARD; *van Ravenzwaaij et al., 2020*) models (see *Figure 1* for an overview). Although the former is a two-sided diffusion process, the latter three models employ a race architecture.

For all models we used the simple delta update rule as a learning model:

$$Q_{i,t+1} = Q_{i,t} + \alpha (r_t - Q_{i,t}) \tag{1}$$

where $Q_{i,t}$ is the value representation of choice option $i$ on trial $t$, $\alpha$ the learning rate, and $r_t$ the reward on trial $t$. The difference between the actual reward and the value representation of the chosen stimulus, $r_t - Q_{i,t}$, is known as the reward prediction error. The learning rate controls the speed

at which Q-values change in response to the reward prediction error, with larger learning rates leading to stronger fluctuations. In this model, only the Q-value of the chosen option is updated.

## RL-EAM 1: RL-DDM

In the first RL-EAM, we use the DDM (*Ratcliff, 1978*) as a choice model (*Figure 1*, left column). The DDM assumes that evidence accumulation is governed by:

$$dx = vdt + sW$$

$v$ is the mean speed of evidence accumulation (the *drift rate*), and $s$ is the standard deviation of the within-trial accumulation white noise (W). The RL-DDM assumes that the drift rate depends linearly on the difference of value representations:

$$v_t = w(Q_{t,1} - Q_{t,2})$$

$w$ is a weighting variable, and $Q_{t,1}$ and $Q_{t,2}$ are the Q-values for both choice options per trial, which change each trial according to *Equation 1*. Hence,

$$dx = w(Q_1 - Q_2)\, dt + sW \tag{2}$$

The starting point of evidence accumulation, $z$, lies between decision boundaries $a$ and $-a$. Here, as in earlier RL-DDM work (*Fontanesi et al., 2019a*; *Fontanesi et al., 2019b*; *Pedersen et al., 2017*), we assume an unbiased start of the decision process (i.e., $z = 0$). Evidence accumulation finishes when threshold $a$ or $-a$ is reached, and the decision for the choice corresponding to $Q_1$ or $Q_2$, respectively, is made. The response time is the time required for the evidence-accumulation process to reach the bound, plus an intercept called the non-decision time ($t0$). The non-decision time is the sum of the time required for perceptual encoding and the time required for the execution of the motor response. Parameter $s$ was fixed to 1 to satisfy scaling constraints (*Donkin et al., 2009*; *van Maanen and Miletić, 2020*). In total, this specification of the RL-DDM has 4 free parameters ($\alpha$, $w$, $a$, $t0$).

Furthermore, we fit four additional RL-DDMs (RL-DDM A1-A4) with between-trial variabilities in start point, drift rate, and non-decision time, as well as a non-linear link function between Q-values and drift rates (*Fontanesi et al., 2019a*). RL-DDM A1 uses the non-linear function $v_t = \frac{2v_{max}}{1+\exp(w(Q_{t,1}-Q_{t,2}))} - v_{max}$ to link Q-values to drift rates (*Fontanesi et al., 2019b*). For the new $v_{max}$ parameter, $\mathfrak{N}(2,5)$ (truncated at 0) and $\Gamma(1,1)$ were used as priors for the hypermean and hyperSD, respectively. RL-DDM A2 includes between-trial variabilities in both drift rate $s_v$ and start point $s_z$, with $\mathfrak{N}(0.1, 0.1)$ and $\mathfrak{N}(0.1, 0.1)$ as priors for hypermeans (respectively, both truncated at 0) and $\Gamma(1,1)$ for the hyperSD. Drift rate variability was estimated as a proportion of the current drift rate, such that $s_{v,t} = v_t * s_v$ (which allows for higher variability terms for higher Q-value differences, but retains the ratio $v/s_v$). RL-DDM A3 included $s_v$, $s_z$, and also between-trial variability in non-decision time $s_{t0}$, for which $\mathfrak{N}(0.1, 0.1)$ (truncated at 0) and $\Gamma(1,1)$ were used as priors for the hypermean and hyperSD, respectively. RL-DDM A4 used all three between-trial variabilities as well as the non-linear link function. The quality of fits of these additional models can be found in *Figure 3—figure supplement 1*. Foreshadowing the results, the RL-DDM A3 improved the quality of fit compared to the RL-DDM, but required an implausibly high non-decision time variability: The across-subject mean of the median posterior estimates of the t0 and $s_{t0}$ parameters indicate a non-decision time distribution of [0.27 s, 0.64 s]. The range of 0.37 s is very high in light of the literature (*Tran et al., 2021*), raising the question of its psychological plausibility. For this reason, as well as since the RL-DDM is used most often without $s_{t0}$, we focus on the RL-DDM (without between-trial variabilities) in the main text.

## RL-EAM 2: RL-RD

The RL-RD (*Figure 1*, middle panel) assumes that two evidence accumulators independently accrue evidence for one choice option each, both racing toward a common threshold $a$ (assuming no response bias). The first accumulator to hit the bound wins, and the corresponding decision is made. For each choice option $i$, the dynamics of accumulation are governed by:

$$dx_i = [V_0 + wQ_i]dt + sW \tag{3}$$

$V_0$ is a parameter specifying the drift rate in the absence of any evidence, $w$ a weighting parameter, and $s$ the standard deviation of within-trial noise. As such, the mean speed of accumulation (the drift rate $v_i$) is the sum of two independent factors: an evidence-independent baseline speed $V_0$, and an evidence-dependent weighted Q-value, $wQ_i$. Since $V_0$ is assumed to be identical across accumulators, and governs the speed of accumulation unrelated to the amount of evidence, we interpret this parameter as an additive urgency signal (**Miletić and van Maanen, 2019**), with conceptually similar behavioral effects as collapsing bounds (**Hawkins et al., 2015**). Similar to the DDM, a non-decision time parameter accounts for the time for perceptual encoding and the motor response time. Parameter $s$ was fixed to 1 to satisfy scaling constraints (**Donkin et al., 2009**; **van Maanen and Miletić, 2020**). In total, the RL-RD has 5 free parameters ($\alpha$, $w$, $a$, $v0$, $t0$).

Each accumulator's first passage times are Wald (also known as inverted Gaussian) distributed (**Anders et al., 2016**). In an independent race model, each accumulator's first passage time distribution is normalized to the probability of the response with which it is associated (**Brown and Heathcote, 2008**; **Turner, 2019**).

## RL-EAM 3 and 4: RL-ARD

Thirdly, we fit two racing diffusion models based on an advantage race architecture (**van Ravenzwaaij et al., 2020**). An advantage race model using an LBA has been shown to provide a natural account for multi-alternative choice phenomena such as Hick's law, as well as stimulus magnitude effects in perceptual decision-making. As in the RL-RD, accumulators race toward a common bound, but the speed of evidence accumulation $v_i$ depends on multiple factors: first, as in the RL-RD, the evidence-independent speed of accumulation $V_0$; second, the *advantage* of the evidence for one choice option over the other (c.f. the DDM, where the difference between evidence for both choice options is accumulated); and third, the *sum* of the total available evidence. Combined, for two accumulators in the RL-EAM framework, this leads to:

$$dx_1 = [V_0 + w_d(Q_1 - Q_2) + w_s(Q_1 + Q_2)]dt + sW$$
$$dx_2 = [V_0 + w_d(Q_2 - Q_1) + w_s(Q_1 + Q_2)]dt + sW \tag{4}$$

In the original work proposing the advantage accumulation framework (**van Ravenzwaaij et al., 2020**), it was shown that the $w_d$ parameter had a much stronger influence on evidence-accumulation rates than the $w_s$ parameter. Therefore, we first fixed the $w_s$ parameter to 0, to test whether the accumulation of *differences* is sufficient to capture all trends in the data. We term this model the RL-lARD (l = limited), which we compare to the RL-ARD in which we fit $w_s$ as a free parameter.

As previously, parameter $s$ was fixed to 1 to satisfy scaling constraints (**Donkin et al., 2009**; **van Maanen and Miletić, 2020**). The RL-ARD also has a threshold, non-decision time, and learning rate parameter, totaling five ($\alpha$, $w_d$, $a$, $V_0$, $t0$) and 6 free parameters ($\alpha$, $w_d$, $w_s$, $a$, $V_0$, $t0$) for the RL-lARD and RL-ARD, respectively. A parameter recovery study (e.g. **Heathcote et al., 2015**; **Miletić et al., 2017**; **Moran, 2016**; **Spektor and Kellen, 2018**) was performed to confirm that data-generating parameters can be recovered using the experimental paradigm at hand. The results are shown in *Figure 3—figure supplement 2*.

## Bayesian hierarchical parameter estimation, posterior predictive distributions, model comparisons

We estimated group-level and subject-level posterior distributions of each model's parameter using a combination of differential evolution (DE) and Markov-chain Monte Carlo sampling (MCMC) with Metropolis-Hastings (**Ter Braak, 2006**; **Turner et al., 2013**). Sampling settings were default as implemented in the Dynamic Models of Choice *R* software (**Heathcote et al., 2019**): The number of chains, D, was three times the number of free parameters. Cross-over probability was set to $2.38/\sqrt{D}$ at the subject level and $U[0, 1]$ at the group level. Migration probability was set to 0.05 during burn-in only. Convergence was assessed using visual inspection of the chain traces and Gelman-Rubin diagnostic (**Brooks and Gelman, 1998**; **Gelman and Rubin, 1992**) (individual and multivariate potential scale factors < 1.03 in all cases).

Hierarchical models were fit assuming independent normal population ("hyper") distributions for each parameter. For all models, we estimated the learning rate on a probit scale (mapping [0, 1] onto the real domain), with a normal prior $\alpha \sim \Phi(\mathfrak{N}(-1.6, 5))$ (**Spektor and Kellen, 2018**). Prior

distributions for all estimated hyper-mean decision-related parameters were vague. RL-EAMs, the threshold parameter $a \sim \mathfrak{N}(3, 5)$ truncated at 0, and $t0 \sim \mathfrak{N}(0.3, 0.5)$ truncated at 0.025 s and 1 s (all estimation was carried out on the seconds scale). For the RL-DDM, $w \sim \mathfrak{N}(2, 5)$. For the RL-RD, $w \sim \mathfrak{N}(9, 5)$, and for the RL-ARD models, $w_d \sim \mathfrak{N}(9, 5)$ and $w_s \sim \mathfrak{N}(0, 3)$. For the hyperSD, a $\Gamma(1, 1)$ distribution was used as prior. Plots of superimposed prior and posterior hyper-distributions confirmed that these prior settings were not influential.

In initial explorations, we also freely estimated the Q-values at trial 0. However, in the RL-EAMs, the posterior distributions for these Q-values consistently converged on 0, which was therefore subsequently used as a fixed value for all results reported in this paper.

To visualize the quality of model fit, we took 100 random samples from the estimated parameter posteriors and simulated the experimental design with these parameters. For each behavioral measure (e.g. RT quantiles, accuracy), credible intervals were estimated by taking the range between the 2.5% and 97.5% quantiles of the averages over participants.

To quantitatively compare the fit of different models, penalized by their complexity, we used the Bayesian predictive information criterion (BPIC; *Ando, 2007*). The BPIC is an analogue of the Bayesian information criterion (BIC), but (unlike the BIC) suitable for models estimated using Bayesian methods. Compared to the deviance information criterion (*Spiegelhalter et al., 2002*), the BPIC penalizes model complexity more strongly to prevent over-fitting (c.f. AIC vs. BIC). Lower BPIC values indicate better trade-offs between fit quality and model complexity.

## Experiment 2

### Participants

23 participants (mean age 19 years old [SD 1.06 years], 7 men, 23 right-handed) were recruited from the subject pool of the Department of Psychology of the University of Amsterdam and participated for course credits. Participants did not participate in the other experiments. All participants had normal or corrected-to-normal vision and gave written informed consent prior to the experiment onset. The study was approved by the local ethics committee.

### Task

Participants performed the same task as in experiment 1, with the addition of an SAT manipulation (*Figure 2C*). The SAT manipulation included both an instructional cue and a response deadline. Prior to each trial, a cue instructed participants to emphasize either decision speed ('SPD') or decision accuracy ('ACC') in the upcoming trial, and in speed trials, participants did not earn points if they were too late (>600 ms). As in experiment 1, after each choice participants received feedback consisting of two components: an outcome and a reward. The outcome refers to the outcome of the probabilistic gamble, whereas the reward refers to the number of points participants actually received. If participants responded in time, the reward was equal to the outcome. In speed trials, participants did not earn points if they responded later than 600 ms after stimulus onset, even if the outcome was +100. On trials where participants responded too late, they were additionally informed of the reward that was associated with their choice, had they been in time. This way, even when participants are too late, they still receive the feedback that can be used to learn from their choices.

The deadline manipulation was added because we hypothesized that instructional cues alone would not be sufficient to persuade participants to change their behavior in the instrumental learning task, since that task specifically requires them to accumulate points. If the received number of points was independent of response times, the optimal strategy to collect most points would be to ignore the cue and focus on accuracy only.

Participants performed 324 trials divided over three blocks. Within each block, three pairs of stimuli were shown, with associated reward probabilities of 0.8/0.2, 0.7/0.3, and 0.6/0.4. Speed and accuracy trials were randomly interleaved. *Figure 2C* depicts the sequence of events in each trial. As this experiment also served as a pilot for an fMRI experiment, we added fixation crosses between each phase of the trial, with jittered durations. A pre-stimulus fixation cross lasted 0.5, 1, 1.5, or 2 s; fixation crosses between cue and stimulus, between stimulus and highlight, and between highlight and feedback lasted 0, 0.5, 1, or 1.5 s; and an inter-trial interval fixation cross lasted 0.5, 1, 1.5, 2, 2.5 s. Each trial took 7.5 s. The experiment took approximately 45 min.

## Exclusion

Four participants did not reach above-chance performance as indicated by a binomial test (cut-off 0.55, p<0.05), and were excluded from further analyses. The final sample thus consisted of 19 participants (mean age 19 years old [SD 1.16 years], 6 men, 19 right-handed). For one additional participant, a technical error occurred after the first block. This participant was included in the analyses, since the Bayesian estimation framework naturally down-weighs the influence of participants with fewer trials.

## Manipulation check and across-block differences in behavior

We expected an interaction between SAT conditions and learning. In the early trials, participants have not yet learned the reward contingencies, causing a low evidence accumulation rate compared to later trials. With low rates it takes longer to reach the decision threshold, and small changes in the threshold settings or drift rates (by means of an additive urgency signal) can cause large behavioral effects. Therefore, we expected the behavioral effects of the SAT manipulation to become smaller over the course of learning.

To formally test for the behavioral effects of the SAT manipulation in experiment 2, we fit two mixed effects models (*Gelman and Hill, 2007*): A linear model with RT as dependent variable, and a logistic model with accuracy as the dependent variable. As fixed effects, trial bin and SAT condition were included. Trial bins were obtained by splitting all trials in ten bins (approximately 20 trials each) per participant. As random effects, only participant was included. For the logistic model using accuracy as a dependent variable, we log-transformed trial bin numbers (*Evans et al., 2018*; *Heathcote et al., 2000*), to account for the non-linear relation between accuracy and trial bin (*Figure 6*, top row). Mixed effects analyses were done using *lme4* (*Bates et al., 2015*). For all mixed effects models, we report parameter estimates of the fixed effects, their standard error and confidence interval, as well as a p-value obtained from a *t*-distribution with the denominator degrees of freedom approximated using Satterthwaite's method (*Satterthwaite, 1941*), as implemented in the *lmerTest* package (*Kuznetsova et al., 2017*) for the *R* programming language (*R Development Core Team, 2017*).

Next, we tested for the across block stability of behavior using two mixed effects models. One linear mixed effects model was used to predict RT with block number, trial bin, and their interaction, with a random intercept for participant. A second, logistic mixed effects model was used to test the effect of block, trial bin (log-transformed, as above), and their interaction on choice accuracy. In both models, trial bin is expected to influence the outcome variables, but the assumption of across block stability in behavior is violated if there are main effects of block number and/or interaction effects between block number and trial bin. Mean RT and accuracy by block is shown together with the formal test results in *Figure 6—figure supplement 3*.

## Cognitive modeling

We fit three RL-DDMs and seven RL-ARDs. The three RL-DDM models varied either threshold, the Q-value weighting on the drift rates parameter (*Sewell and Stallman, 2020*), or both. The seven RL-ARD allowed all unique combinations of the threshold, urgency, and drift rate parameters free to vary between the speed and accuracy conditions.

For the accuracy condition, we used the same priors as in experiment 1. In the speed condition, the parameters that were free to vary were estimated as proportional differences from the accuracy conditions; specifically: $a_{spd} = (1 + m_{a,spd}) * a_{acc}$, $V_{0,spd} = (1 + m_{V_0,spd}) * V_{0,acc}$, and $v_{i,spd} = (1 + m_{v,spd}) * v_{i,acc}$. The prior used was $\mathfrak{N}(0, 5)$ for the hypermean and $\Gamma(1, 1)$ for the hyperSD of all parameters $m$, truncated at -1.

As in experiment 1, we performed a parameter recovery study to confirm that the data-generating parameters can be recovered, using the winning model and a simulation of the paradigm of experiment 2. The results are shown in *Figure 6—figure supplement 2*.

Additionally, we performed a second model comparison using three variants of RL-DDM A3 (i.e., including between-trial variabilities in start point, drift rate, and non-decision time), which varied the threshold, the Q-value weighting on the drift rate parameters, and both. The results are shown in *Figure 6—figure supplement 1*, and lead to the same conclusions as the RL-DDM.

## Experiment 3

### Participants

Forty-seven participants (mean age 21 years old [SD 2.81 years], 16 men, 40 right-handed) were recruited from the subject pool of the Department of Psychology of the University of Amsterdam and participated for course credits. Participants did not participate in the other experiments. All participants had normal or corrected-to-normal vision and gave written informed consent prior to the experiment onset. The study was approved by the local ethics committee.

### Task

The reversal learning task had the same general task structure as experiment 1. Participants completed four blocks of 128 trials each, totaling 512 trials. Within each block, two pairs of stimuli were randomly interleaved, with associated reward probabilities of 0.8/0.2 and 0.7/0.3. Between trials 61 and 68 (uniformly sampled) of each block, the reward probability switched between stimuli, such that the stimulus with a pre-reversal reward probability of 0.8/0.7 had a post-reversal reward probability of 0.2/0.3 (and vice versa). Participants were not informed of the reversals prior to the experiment, but many reported noticing them.

In addition to the reversal learning task, the experimental session also contained a working memory task that is not of current interest. Thirty participants performed the reversal learning task before the working memory task, and 17 participants afterwards. The entire experiment took approximately one hour.

### Cognitive modeling

We first tested whether a standard soft-max model is able to capture the quick change in choice behavior after the occurrence of a reversal. Soft-max is given by:

$$P_{i,t} = \frac{\exp \beta Q_{i,t}}{\sum\limits_{j}^{J} \exp \beta Q_{i,t}} \tag{5}$$

where $P_{i,t}$ is the probability of choosing option $i$ on trial $t$, $J$ is the total number of choice options, and $\beta$ is a free parameter often called the inverse temperature. The inverse temperature is often interpreted in terms of the exploration/exploitation trade-off (*Daw et al., 2006*), with higher values indicating more exploitation. In two-choice settings, *Equation* 5 can be re-written as:

$$P_{2,t} = \frac{1}{1 + \exp \beta (Q_{1,t} - Q_{2,t})} \tag{6}$$

which highlights that the choice probability is driven by the *difference* in Q-values, weighted by the inverse temperature parameter. We hierarchically fit the soft-max model using the same parameter estimation methods as in experiment 1. Priors for the hypermean were set to $\beta \sim N(1,5)$ truncated at 0, and for the hyperSD $\Gamma(1,1)$.

Then, the RL-DDM and RL-ARD were fit to the data using the same methods as in experiment 1. Again, we performed a parameter recovery study, of which the results are shown in *Figure 7—figure supplement 3*. Similarly, we also fit RL-DDM A3 to the data. In an initial fit, the MCMC chains for 11 (out of 47) participants got stuck in values for $s_z$ of 1 (i.e. $s_z$ covered the entire range between both thresholds), which are implausibly high and moreover led to convergence problems. We re-fit this model with the prior on $s_z \sim \mathfrak{N}(0.1,0.1)$ truncated at 0 and 0.5 (i.e. setting the maximum range of between-trial start point variability to be half the range between the lower and upper threshold), which did converge. The posterior predictives are shown in *Figure 7—figure supplement 2*. This model led to the same overall conclusions as the standard RL-DDM.

## Experiment 4

### Participants

Forty-three participants (mean age 20 years old [SD 4.29 years], 5 men, 36 right-handed) were recruited from the subject pool of the Department of Psychology of the University of Amsterdam and participated for course credits. They did not participate in the other experiments. All

participants had normal or corrected-to-normal vision and gave written informed consent prior to the experiment onset. The study was approved by the local ethics committee. Participants performed the task online.

## Task

The three-alternative instrumental learning task had the same general task structure as before. Participants completed three blocks of 144 trials each, totaling 432 trials. Within each block, four triplets of stimuli were randomly interleaved. We used a factorial design to manipulate both difficulty (defined as the difference between the target and distractor reward probability) and reward magnitude (defined as the mean probability of reward per triplet): 0.8/0.25/0.25 (easy, high magnitude), 0.7/0.3/0.3 (hard, high magnitude), 0.7/0.15/0.15 (easy, low magnitude), and 0.6/0.2/0.2 (hard, low magnitude). The entire experiment took approximately 30 min.

## Exclusion

Nine subjects did not reach above-chance accuracy and were excluded from analysis (accuracy cut-off 0.37, corresponding to $p<0.05$ in a binomial test). The final sample thus consisted of 34 subjects (three men, mean age 20 years old [SD 4.68 years], 28 right-handed).

## Cognitive modeling

A three-alternative RL-ARD with the Win-All stopping rule was fit to the data. The multi-alternative version of the RL-ARD includes one accumulator per directional pairwise difference between choice options, leading to a total of six accumulators (1-2, 1-3, 2-1, 2-3, 3-1, 3-2) for the case of three choice options. Each accumulator is governed by the same evidence-accumulation dynamics as in the two-alternative RL-ARD:

$$dx_{1-2} = [V_0 + w_d(Q_1 - Q_2) + w_s(Q_1 + Q_2)]dt + sW$$
$$dx_{1-3} = [V_0 + w_d(Q_1 - Q_3) + w_s(Q_1 + Q_3)]dt + sW$$

$$dx_{2-1} = [V_0 + w_d(Q_2 - Q_1) + w_s(Q_2 + Q_1)]dt + sW$$
$$dx_{2-3} = [V_0 + w_d(Q_2 - Q_3) + w_s(Q_2 + Q_3)]dt + sW \qquad (7)$$

$$dx_{3-1} = [V_0 + w_d(Q_3 - Q_1) + w_s(Q_3 + Q_1)]dt + sW$$
$$dx_{3-2} = [V_0 + w_d(Q_3 - Q_2) + w_s(Q_3 + Q_2)]dt + sW$$

Hence, each choice option (e.g. 1) is associated with two accumulators that collect evidence for the *advantage* of that option over the other two options (1-2, 1-3). The Win-All stopping rule proposes that a final choice is made when two conditions are satisfied: (1) *both* accumulators corresponding to that choice reached their thresholds, *and* (2) for each other choice option *at least one* accumulator has not yet reached the threshold. The corresponding response time is the decision time of the slowest of the two winning accumulators, plus the non-decision time. The probability of response one at time *t* is then given by *van Ravenzwaaij et al., 2020*; derivation in their Appendix:

$$p_1(t) = \sum_{I \neq 1}\left[ PDF_{1-I}(t) \times \prod_{J \neq [1,I]} CDF_{1-J}(t) \right] \times \prod_{I \neq 1}\left[ 1 - \prod_{K \neq I} CDF_{I-K}(t) \right]$$

where I are choice options {2,3}, J is an option in {2,3} that is not I, and K is an option in {1,2,3} that is not I. PDF and CDF refer to the probability density and cumulative density function of the Wald distribution.

Despite the visual complexity of *Figure 9*, the three-alternative RL-ARD remains highly constrained by the linking functions of *Equation 7* and the Q-value evolution. As only the architecture of the linking between Q-values and accumulators was generalized to accommodate the third choice option, there remains the same number of parameters as in the two-alternative case. Note that the three-alternative Win-All RL-ARD naturally reduces to the two-choice RL-ARD when one of the choice options is removed.

We fit the three-alternative RL-ARD using the same fitting methods (including the same priors) as in experiment 1. We also performed a parameter recovery study using the same methods as in earlier experiments, the results of which are shown in *Figure 11—figure supplement 1*.

### Code availability statement
All analysis codes are available on OSF (https://osf.io/ygrve/).

## Acknowledgements

We thank Barbara Mathiopoulou and Chris Riddell for their help collecting the data. This work was supported by an NWO-VICI grant (BUF), an ABC VIP grant and ARC DP150100272 and DP160101891 grants (AH).

## Additional information

### Competing interests
Birte U Forstmann: Reviewing editor, *eLife*. The other authors declare that no competing interests exist.

### Funding

| Funder | Grant reference number | Author |
| --- | --- | --- |
| Nederlandse Organisatie voor Wetenschappelijk Onderzoek | 016.vici.185.052 | Birte U Forstmann |
| Australian Research Council | DP150100272 | Andrew Heathcote |
| Australian Research Council | DP160101891 | Andrew Heathcote |
| University of Amsterdam | VIP Grant | Andrew Heathcote |

The funders had no role in study design, data collection and interpretation, or the decision to submit the work for publication.

### Author contributions
Steven Miletić, Conceptualization, Resources, Data curation, Software, Formal analysis, Validation, Investigation, Visualization, Methodology, Writing - original draft, Writing - review and editing; Russell J Boag, Conceptualization, Formal analysis, Investigation, Methodology, Writing - review and editing; Anne C Trutti, Resources, Data curation, Investigation, Writing - review and editing; Niek Stevenson, Conceptualization, Data curation, Investigation, Methodology, Writing - review and editing; Birte U Forstmann, Conceptualization, Resources, Supervision, Funding acquisition, Project administration, Writing - review and editing; Andrew Heathcote, Conceptualization, Resources, Software, Formal analysis, Supervision, Funding acquisition, Validation, Investigation, Methodology, Writing - original draft, Writing - review and editing

### Author ORCIDs
Steven Miletić https://orcid.org/0000-0001-7399-2926
Russell J Boag https://orcid.org/0000-0002-7689-0682
Anne C Trutti https://orcid.org/0000-0002-0044-4846
Niek Stevenson https://orcid.org/0000-0003-3206-7544
Birte U Forstmann http://orcid.org/0000-0002-1005-1675
Andrew Heathcote https://orcid.org/0000-0003-4324-5537

### Ethics
Human subjects: Informed consent was obtained in all experiments prior to the experiment onset. The local ethics board of the Department of Psychology, University of Amsterdam, approved the

study, with reference numbers 2018-BC-9620 (experiment 1), 2019-BC-10672 (experiment 2), 2019-BC-10250 (experiment 3), and 2020-BC-12788 (experiment 4).

## Decision letter and Author response

Decision letter https://doi.org/10.7554/eLife.63055.sa1
Author response https://doi.org/10.7554/eLife.63055.sa2

---

# Additional files

## Supplementary files

• Transparent reporting form

## Data availability

All data analysed in this study are available from https://osf.io/ygrve/.

The following dataset was generated:

| Author(s) | Year | Dataset title | Dataset URL | Database and Identifier |
|---|---|---|---|---|
| Miletic S, Boag RJ, Trutti AC, Stevenson N, Forstmann BU, Heathcote A | 2020 | A new model of decision processing in instrumental learning tasks | https://osf.io/ygrve/ | Open Science Framework, ygrve |

---

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
