## [Decision Letter]

**Acceptance summary:**

Your responses to the reviewers' comments have convincingly addressed earlier concerns as to how the advantage racing model fares in reinforcement problems with more than two alternatives – a situation where its superiority over standard drift-diffusion models is clearest. The additional experiment described in the revised manuscript, and the formal derivation of the advantage racing model for reinforcement problems with more than two alternatives, constitute two important additions to the study which now provides a qualitative theoretical advance over the existing literature. The revised manuscript describes a convincing set of experiments and analyses that supports advantage racing as a good model of the interplay between learning and decision-making in reinforcement problems varying stimulus difficulty, speed-accuracy trade-off and stimulus-response mapping. Congratulations for an insightful study of human behavior.

**Decision letter after peer review:**

Thank you for submitting your article "A new model of decision processing in instrumental learning tasks" for consideration by *eLife*. Your article has been reviewed by two peer reviewers, and the evaluation has been overseen by a Reviewing Editor and Joshua Gold as the Senior Editor. The following individual involved in review of your submission has agreed to reveal their identity: Jan Willem de Gee (Reviewer #1).

The reviewers have discussed the reviews with one another and the Reviewing Editor has drafted this decision to help you prepare a revised submission.

As the editors have judged that your manuscript is of interest, but as described below that additional experiments are required before it is published, we would like to draw your attention to changes in our revision policy that we have made in response to COVID-19 (https://elifesciences.org/articles/57162). First, because many researchers have temporarily lost access to the labs, we will give authors as much time as they need to submit revised manuscripts. We are also offering, if you choose, to post the manuscript to bioRxiv (if it is not already there) along with this decision letter and a formal designation that the manuscript is "in revision at *eLife*". Please let us know if you would like to pursue this option.

Summary:

This cognitive modeling study on a timely topic investigates the combination of reinforcement learning and decision-making for modeling choice and reaction-time data in sequential reinforcement problems (e.g., bandit tasks). The central claim of the paper is that the often-used combination of reinforcement learning with the drift-diffusion model (which decides based on the difference between option values) does not provide an adequate model of instrumental learning. Instead, the authors propose an "advantage racing" model which provides better fits to choice and reaction-time data in different variants of two-alternative forced-choice tasks. Furthermore, the authors emphasize that their advantage racing model allows for fitting decision problems with more than two alternatives – something which the standard drift-diffusion model cannot do. These findings can be of interest for researchers investigating learning and decision-making.

The study asks an important question for understanding the interaction between reinforcement learning and decision-making, the methods appear sound, and the manuscript is clearly written. The superiority of the advantage racing model is key to the novelty of the study, which otherwise relies on a canonical task studied in several recent papers on the same issue. However, the reviewers feel that the framing of the study and its conclusions would require additional analyses experiments to transform the manuscript from a modest quantitative improvement into a qualitative theoretical advance. In particular, as described in the paragraphs below, the authors should test how their advantage racing model fares in reinforcement problems with more than two alternatives. This is, from their own account throughout the paper, a situation where their model could show most clearly its superiority over standard drift-diffusion models used in the recent literature. The section below describes the essential revisions that the authors should address in a point-by-point fashion in a revised version of their manuscript.

Essential revisions:

1) Starting from the Abstract, the authors allude to reinforcement problems with more than 2 alternatives (1) and to the role of absolute choice values (2) in characterizing the limitations of the standard DDM. However, in the current manuscript, point 1 is not addressed (and actually does not appear to be amenable to the current model implementation, see the next point), and point 2 is addressed via modest quantitative improvements to model fits rather than true qualitative divergence between their model and other models' ability to capture specific behavioral effects.

The authors could greatly strengthen their conclusions if they extend their model to RL data sets with (1) more than two alternatives, and (2) variations in the absolute mean reward across blocks of learning trials. These experimental variants, which would put the superiority of the advantage racing model to a real test, would afford to test qualitative predictions possibly not shared by existing alternatives. For instance, does their model predict set size effects during instrumental learning? Does their model predict qualitative shifts in choice and reaction times when different task blocks have different mean rewards? At the moment the primary results rely solely on improved fits, but it would be important to show their model's unique ability to capture more salient qualitative behavioral effects. This point requires either the reanalysis of existing data from reinforcement problems with more than two alternatives and/or with block-wise variations in mean reward, or the collection of additional data. We understand the difficulty of collecting data in the current context, but this additional data is very important to support the qualitative superiority of the advantage racing model.

2) It is not clear how the advantage racing model would easily transfer to problems with more than two alternatives. As depicted in Equation 1, the model depends on the difference between two unique Q-values (weighted by *w_D_*). How would this be implemented with more than two alternatives? There are several possibilities of implementations (e.g., the current Q-value relative to the top Q-value, the current Q-value minus the average Q-value, etc.), but they seem to require additional heuristics that are not described in the current implementation of the advantage racing model. The authors could also incorporate modulation of drift rates by policies, or use an actor-critic approach. In any case, the model in its current form cannot straightforwardly transfer to more than two alternatives. Given that the authors stress the benefits of their model for sequential reinforcement problems with more than two alternatives, they should provide an implementation of their model that can transfer to more than two alternatives (and also test this implementation against human data in such problems, see the previous point).

3) All three racing models (including the authors' preferred model) implement an urgency signal. It is unclear why the authors did not consider a similar mechanism within the DDM framework. Within this framework, urgency could possibly be implemented either as (linearly or hyperbolically) collapsing bounds, or as self-excitation (inverse of leak). Both implementations require only one extra parameter, as in the racing models used by the authors. The lack of explicit urgency needs to be addressed, because the current superiority of the advantage racing model could be due in part to its urgency component which is absent from the drift-diffusion model. And the importance of urgency in such problems has already been reported in earlier studies.

4) The authors appreciate the rigorous parameter recovery analyses in the supplement, but it would be very useful to perform also a model “separability” analysis – i.e., plot a confusion matrix between alternative models. Indeed, it seems like several of the tested models are relatively similar qualitatively speaking and it would thus be very useful to know how confusable they are. The fact that parameters from one known ground-truth model can be recovered accurately does not speak to the confusability between different models.

5) It is unclear that the authors are implementing SARSA (Rummery and Niranjan, 1994). Indeed, SARSA is: Q(s,a)[t+1] = Q(s,a)[t] + lr(r[t+1] + discount(Q(s,a)[t+1]) – Q(s,a)[t]. However, this task is a “single-step” problem. So it seems like the authors are rather implementing SAR – i.e., the standard Rescorla-Wagner “delta” rule. Please clarify this modeling aspect.

6) The basic aggregate behavior and model fitting quality should be described and illustrated beyond percentile distributions of reaction-times for correct and error trials. The authors should plot the grand-average RT distributions for subjects and best-fitting model predictions (pooled across subjects and trials). How variable were subjects' behavior? The reviewers understand that the model was fit hierarchically, but it would be nice (possibly in the supplement) to see a distribution of fit quality across subjects (1), to see RT distributions of a couple of good and bad fits (2), and to check whether the results hold after excluding the subjects with worst fits (if there are any outliers, 3). Related, in the RT percentile plots (Figures 3 and 4), it would be useful to see some measure of variability between subjects.

7) The authors write that RL-AEMs assume that "[...] a subject gradually accumulates evidence for each choice option by sampling from a distribution of memory representations of the subjective value (or expected reward) associated with each choice option (known as Q-values)." It is unclear what the authors mean by "a distribution of memory representations". Either the authors could replace this by "a running average of rewards", or they truly mean something like sampling from memory – like recent papers e.g., Bornstein et al., 2017, Nat. Comm. Sampling from (a distribution of) memory representations is a relatively new idea, and I think it would help if the authors would be more circumscribed in the interpretation of these results, and also provide more context and rationale both in the Introduction and Discussion. For example, an interesting Discussion paragraph would be on how such a memory-sampling process might actually be implemented in the brain.

---

## [Author Response]

Essential revisions:1) Starting from the Abstract, the authors allude to reinforcement problems with more than 2 alternatives (1) and to the role of absolute choice values (2) in characterizing the limitations of the standard DDM. However, in the current manuscript, point 1 is not addressed (and actually does not appear to be amenable to the current model implementation, see the next point), and point 2 is addressed via modest quantitative improvements to model fits rather than true qualitative divergence between their model and other models' ability to capture specific behavioral effects. The authors could greatly strengthen their conclusions if they extend their model to RL data sets with (1) more than two alternatives, and (2) variations in the absolute mean reward across blocks of learning trials. These experimental variants, which would put the superiority of the advantage racing model to a real test, would afford to test qualitative predictions possibly not shared by existing alternatives. For instance, does their model predict set size effects during instrumental learning? Does their model predict qualitative shifts in choice and reaction times when different task blocks have different mean rewards? At the moment the primary results rely solely on improved fits, but it would be important to show their model's unique ability to capture more salient qualitative behavioral effects. This point requires either the reanalysis of existing data from reinforcement problems with more than two alternatives and/or with block-wise variations in mean reward, or the collection of additional data. We understand the difficulty of collecting data in the current context, but this additional data is very important to support the qualitative superiority of the advantage racing model.

We followed the reviewers’ advice to perform an additional experiment. Specifically, we designed an instrumental learning task in which the participants had to repeatedly choose between three choice alternatives. The RL-DDM cannot be applied in settings beyond two-choice alternatives, providing a strong qualitative divergence between the models’ predictions and capabilities. Within this multi-alternative experiment, we included a reward magnitude manipulation to test whether the RL-ARD is able to capture the behavioral effects of changes in overall reward magnitude.

We describe the new experiment as well as the multi-alternative RL-ARD in the main text of the revised manuscript. For convenience, we copied the relevant text below:

**“**Multi-alternative choice

Finally, we again drew on Van Ravenzwaaij et al.’s (Van Ravenzwaaij et al., 2020) advantage framework to extend the RL-ARD to multi-alternative choice tasks, a domain where the RL-DDM cannot be applied. […] As a consequence, *both* the Q-value sum (weighted by median wS=0.15), and the smaller changes in the Q-value difference (weighted by median wD=1.6), increased the drift rates for the response accumulators (vT−D; Figure 12D), which led to higher accuracy and faster responses.”

Furthermore, the Materials and methods section provides more detail on experiment 4 and on the implementation of the RL-ARD:

**“**Experiment 4

Participants

43 participants (mean age 20 years old [SD 4.29 years], 5 men, 36 right-handed) were recruited from the subject pool of the Department of Psychology of the University of Amsterdam and participated for course credits. […] We also performed a parameter recovery study using the same methods as in earlier experiments, the results of which are shown in Figure 11—figure supplement 1.”

2) It is not clear how the advantage racing model would easily transfer to problems with more than two alternatives. As depicted in Equation 1, the model depends on the difference between two unique Q-values (weighted by w_D_). How would this be implemented with more than two alternatives? There are several possibilities of implementations (e.g., the current Q-value relative to the top Q-value, the current Q-value minus the average Q-value, etc.), but they seem to require additional heuristics that are not described in the current implementation of the advantage racing model. The authors could also incorporate modulation of drift rates by policies, or use an actor-critic approach. In any case, the model in its current form cannot straightforwardly transfer to more than two alternatives. Given that the authors stress the benefits of their model for sequential reinforcement problems with more than two alternatives, they should provide an implementation of their model that can transfer to more than two alternatives (and also test this implementation against human data in such problems, see the previous point).

We addressed the implementation of the multi-alternative RL-ARD in response to comment 1 above.

3) All three racing models (including the authors' preferred model) implement an urgency signal. It is unclear why the authors did not consider a similar mechanism within the DDM framework. Within this framework, urgency could possibly be implemented either as (linearly or hyperbolically) collapsing bounds, or as self-excitation (inverse of leak). Both implementations require only one extra parameter, as in the racing models used by the authors. The lack of explicit urgency needs to be addressed, because the current superiority of the advantage racing model could be due in part to its urgency component which is absent from the drift-diffusion model. And the importance of urgency in such problems has already been reported in earlier studies.

We agree with the reviewer that including an urgency mechanism in the RL-DDM could potentially improve the model’s quality of fit. However, there is no known analytical solution to the likelihood of the DDM with collapsing bounds or an urgency gain signal. Researchers interested in fitting the DDM with an urgency mechanism therefore must resort to simulation-based approximations of the likelihood function. In stationary experiments, the urgency-DDM must be simulated for *N* trials (commonly in the range of 10,000-50,000; e.g., Miletić et al., 2017; Trueblood et al., 2020) per condition per subject in each iteration of the MCMC sampling process. While computationally expensive, with good computational facilities this can still be feasible. However, the RL-DDM has single-trial drift rates, implying that the likelihood distribution needs to be approximated for each individual trial separately, hence requiring the simulation of *N* trials *per trial* per subject, providing a 208-fold increase in the number of trials that need to be simulated for experiment 1 alone (and more for the other experiments). The explosion in the number of simulations becomes computationally infeasible.

Secondly, it is important to point out that the origin of the concept of urgency lies in racing accumulator models, similar to the implementation we used in the RL-ARD. The development of DDMs with collapsing bounds and urgency gain signals was only later. As such, our model’s urgency mechanism remains close to the original concept.

In the revised manuscript, we address this point in the Discussion:

“Although an urgency mechanism such as collapsing bounds (Boehm et al., 2016; Bowman et al., 2012; Hawkins et al., 2015; Milosavljevic et al., 2010) or gain modulation (Boehm et al., 2016; Churchland et al., 2008; Ditterich, 2006; Hawkins et al., 2015) could potentially improve the fits of the RL-DDM, fitting DDMs with such mechanisms is computationally very expensive, usually requiring the researcher to use simulation-based approximations (e.g., Turner and Sederberg, 2014). […] Furthermore, the origin of the concept of urgency lies in studies using racing accumulator models (Ditterich, 2006; Mazurek et al., 2003; Reddi and Carpenter, 2000), which was only later incorporated in the DDM (Milosavljevic et al., 2010); the implementation in the RL-ARD remains conceptually close to the early proposals.”

4) The authors appreciate the rigorous parameter recovery analyses in the supplement, but it would be very useful to perform also a model “separability” analysis – i.e., plot a confusion matrix between alternative models. Indeed, it seems like several of the tested models are relatively similar qualitatively speaking and it would thus be very useful to know how confusable they are. The fact that parameters from one known ground-truth model can be recovered accurately does not speak to the confusability between different models.

In the revised manuscript, we included confusion matrices as Figure 3—figure supplement 3. Since our main focus is on the comparison between the RL-ARD and the RL-DDM (and the generation of the confusion matrices is computationally intensive), we only compared these two models. These results show that the BPIC nearly always correctly identified the data-generating models.

5) It is unclear that the authors are implementing SARSA (Rummery and Niranjan, 1994). Indeed, SARSA is: Q(s,a)[t+1] = Q(s,a)[t] + lr(r[t+1] + discount(Q(s,a)[t+1]) – Q(s,a)[t]. However, this task is a “single-step” problem. So it seems like the authors are rather implementing SAR – i.e., the standard Rescorla-Wagner “delta” rule. Please clarify this modeling aspect.

We apologize for this confusion, and removed all references to “SARSA”, which we substituted with the “delta rule”.

6) The basic aggregate behavior and model fitting quality should be described and illustrated beyond percentile distributions of reaction-times for correct and error trials. The authors should plot the grand-average RT distributions for subjects and best-fitting model predictions (pooled across subjects and trials). How variable were subjects' behavior? The reviewers understand that the model was fit hierarchically, but it would be nice (possibly in the supplement) to see a distribution of fit quality across subjects (1), to see RT distributions of a couple of good and bad fits (2), and to check whether the results hold after excluding the subjects with worst fits (if there are any outliers, 3). Related, in the RT percentile plots (Figures 3 and 4), it would be useful to see some measure of variability between subjects.

We included the following additional plots:

Figure 3—figure supplement 4, showing grand average RT distributions (both data and posterior predictive distributions), as well as the RT distributions for the first 10 subjects. From visual inspection of similar plots for all subjects, we found no subjects for which the model clearly misfit the empirical RT distributions; the quality of fit of the first 10 subjects was representative of the quality of fit of all other subjects. For experiments 2-4, similar plots are now provided as Figure 6—figure supplement 4, Figure 7—figure supplement 4, Figure 11—figure supplement 2, respectively.

Figure 4—figure supplement 4, which replicates Figure 4 but includes error bars (standard error) on points of empirical data to visualize between-subject variability. Since this plot combines two different approaches to the concept of uncertainty (a frequentist approach, where uncertainty is assumed to lie in the data, and a Bayesian approach, where the uncertainty is assumed to lie in the model), we prefer to keep this figure supplementary instead of replacing the main figure to retain consistency in our Bayesian approach in the main text. For experiments 2-4, similar plots are now provided as Figure 6—figure supplement 5, Figure 7—figure supplement 5, and Figure 11—figure supplement 3, respectively.

7) The authors write that RL-AEMs assume that "[…] a subject gradually accumulates evidence for each choice option by sampling from a distribution of memory representations of the subjective value (or expected reward) associated with each choice option (known as Q-values)." It is unclear what the authors mean by "a distribution of memory representations". Either the authors could replace this by "a running average of rewards", or they truly mean something like sampling from memory – like recent papers e.g., Bornstein et al., 2017, Nat. Comm. Sampling from (a distribution of) memory representations is a relatively new idea, and I think it would help if the authors would be more circumscribed in the interpretation of these results, and also provide more context and rationale both in the Introduction and Discussion. For example, an interesting Discussion paragraph would be on how such a memory-sampling process might actually be implemented in the brain.

We apologize for the confusion raised by our original phrasing. We intended that drift rates are sampled from a running average of rewards. We rephrased the sentence accordingly.